Corrected: Author correction

# Parallels between experimental and natural evolution of legume symbionts

Camille Clerissi[1,2,3], Marie Touchon[1,2], Delphine Capela  [3], Mingxing Tang[3], Stéphane Cruveiller[4], Clémence Genthon[5], Céline Lopez-Roques[5], Matthew A. Parker[6], Lionel Moulin[7], Catherine Masson-Boivin[3] & Eduardo P.C. Rocha [1,2]

The emergence of symbiotic interactions has been studied using population genomics in nature and experimental evolution in the laboratory, but the parallels between these processes remain unknown. Here we compare the emergence of rhizobia after the horizontal transfer of a symbiotic plasmid in natural populations of *Cupriavidus taiwanensis*, over 10 MY ago, with the experimental evolution of symbiotic *Ralstonia solanacearum* for a few hundred generations. In spite of major differences in terms of time span, environment, genetic background, and phenotypic achievement, both processes resulted in rapid genetic diversification dominated by purifying selection. We observe no adaptation in the plasmid carrying the genes responsible for the ecological transition. Instead, adaptation was associated with positive selection in a set of genes that led to the co-option of the same quorum-sensing system in both processes. Our results provide evidence for similarities in experimental and natural evolutionary transitions and highlight the potential of comparisons between both processes to understand symbiogenesis.

[1] Microbial Evolutionary Genomics, Institut Pasteur, 28 rue Dr. Roux, 75015 Paris, France. [2] UMR3525, CNRS, 28 rue Dr. Roux, 75015 Paris, France. [3] LIPM, Université de Toulouse, INRA, CNRS, 31326 Castanet-Tolosan, France. [4] LABGeM, Génomique Métabolique, Genoscope, Institut François Jacob, CEA, CNRS, Université d'Evry, Université Paris-Saclay, 2 rue Gaston Crémieux, 91057 Evry, France. [5] INRA, US 1426, GeT-PlaGe, Genotoul, 31326, Castanet-Tolosan, France. [6] Department of Biological Sciences, State University of New York, 4400 Vestal Parkway East, PO Box 6000, Binghamton, NY 13902, USA. [7] IRD, Cirad, Université de Montpellier, IPME, 911, avenue Agropolis—BP64501, 34394 Montpellier Cedex 5, France. Correspondence and requests for materials should be addressed to C.M-B. (email: Catherine.masson@inra.fr) or to E.P.C.R. (email: erocha@pasteur.fr)

Biological adaptations can be studied using genomic or phenotypic comparisons of natural isolates, including fossil records when they are available, as well as experimental and population analyses of fitness variation. Recently, these approaches have been increasingly complemented by experimental evolution studies. The latter can be done on controlled environments and provide nearly complete "fossil" records of past events because individuals from intermediate points in the experiment can be kept for later analysis[1,2]. Sequencing and phenotyping of evolved clones provides crucial information on the mechanisms driving adaptation in simplified environments. Yet, there are little data on the adaptation of lineages when the process is complex (requires numerous steps). There is even less data on how these experiences recapitulate natural processes (but see refs. [3,4]), raising doubts on the applicability and relevance of experimental evolution studies to understand natural history[5].

Many descriptions of adaptations involving ecological transitions towards pathogenic or mutualistic symbiosis include an initial acquisition via horizontal transfer of genes that provide novel functionalities[6]. For example, the extreme virulence of *Shigella* spp., *Yersinia* spp., or *Bacillus anthracis* results from the acquisition of plasmid-encoded virulence factors by otherwise poorly virulent clones[7–9]. Adaptation is often coupled with the genetic rewiring of the recipient genome, a process that may take hundreds to millions of years in natura[10], and may require specific genetic backgrounds[11]. A striking case of transition mediated by horizontal gene transfer towards mutualism concerns the rhizobium–legume symbiosis, a symbiosis of major ecological importance that contributes to ca. 25% of the global nitrogen cycling. Rhizobia induce the formation of new organs, the nodules, on the root of legumes, which they colonize intracellularly and in which they fix nitrogen to the benefit of the plant[12]. These symbiotic capacities emerged several times in the natural history of α- and β-Proteobacteria, from the horizontal transfer of the key symbiotic genes into soil free-living bacteria (i.e., the *nod* genes for organ formation and the *nif/fix* genes for nitrogen fixation), and were further shaped under plant selection pressure[13–15]. Indeed, legumes have developed control mechanisms that allow the selection of most compatible and beneficial symbionts[16]. There are now hundreds of known rhizobial species scattered in 14 known genera, including the genus *Cupriavidus* in β-proteobacteria[17].

Transition towards legume symbiosis has recently been tested at the laboratory time-scale using an experimental system[18]. A plant pathogen was evolved to become a legume symbiont by mimicking the natural evolution of rhizobia at an accelerated pace. First, the plasmid pRalta[LMG19424]—encoding the key genes allowing the symbiosis between *Cupriavidus taiwanensis* LMG19424[19] and *Mimosa*—was introduced into *Ralstonia solanacearum* GMI1000. The resulting chimera was further evolved under *Mimosa pudica* selective pressure. The chimeric ancestor, which was strictly extracellular and pathogenic on *Arabidopsis thaliana*—but not on *M. pudica* and unable to nodulate it—progressively adapted to become a legume symbiont during serial cycles of inoculation to the plant and subsequent re-isolation from nodules[18,20,21]. Several adaptive mutations driving acquisition and/or drastic improvement of nodulation and infection were previously identified[18,22,23]. Lab-evolution was accelerated by stress-responsive error-prone DNA polymerases encoded in the plasmid which increased the mutation load ex planta[24].

Here, we trace the natural evolutionary history of *C. taiwanensis*, a *Mimosa* rhizobium, and compare it to the experimental evolution of *Ralstonia* into *M. pudica* symbionts, using population genomics and functional enrichment analyses. We specifically focused on patterns of evolution that were previously highlighted by experimental evolution: accumulation of genetic diversity, general patterns of natural selection, chromosomal vs. plasmid adaptation, and evolution of orthologous genes implicated in symbiotic adaptation (type III secretion system, global regulators, mutagenic cassette). We provide evidence that, despite fundamental differences in terms of time frame, protagonists, environmental context, and symbiosis achievement, there were significant parallels in the two processes.

## Results

**Diversification of experimentally evolved *Mimosa* symbionts.** We previously generated 18 independent symbiotic lineages of the *R. solanacearum* GMI1000-pRalta[LMG19424] chimeras that nodulate *M. pudica*[20]. Each lineage was subject to 16 successive cycles of evolution in presence of the plant. We isolated one clone in each of the lineages after the final cycle to identify its genetic and phenotypic differences relative to the ancestor (Supplementary Data 1). The symbiotic performances of the evolved clones improved in the experiment with wide variations between lineages. Fifteen out of the 18 final clones were able to induce the formation of intracellularly infected nodules (Fig. 1a). Yet, none of them fixed nitrogen to the benefit of the plant at this stage. In addition to a total of ca. 1200 point mutations relative to the ancestral clones[20], we detected several large deletions in all clones (Fig. 1a).

Convergent evolution has been observed in previous evolution experiments[25,26]. Thus, we first identified the parallels between the evolved clones for SNP, indels and large deletions. Almost all genetic deletions occurred in homologous regions of the symbiotic plasmid and were systematically flanked by transposable elements that probably mediated their loss by recombination (Supplementary Table 1). These regions had almost only genes of unknown function. Point mutations showed fewer parallelisms. Out of 1147 positions identified as mutated in the final clones, only 12 were found at the same nucleotide position in more than one clone (Fig. 1b). Even if these positions were rare, they were observed (O) more frequently than expected (E) by chance $((O − E)/(O + E) = 0.98$, $P = 0.01$, test based on simulated mutations, see Methods). We then aggregated intragenic mutations per gene and found that the number of genes with mutations in more than one clone was slightly larger than expected by chance $((O − E)/(O + E) = 0.10$, $P = 0.02$, same test). Distribution of point mutations present in more than one lineage was also not random in terms of COG functions $((O − E)/(O + E) = 0.23$, $P = 0.01$, same test). Similarly to previous studies[25,26], this analysis highlights that parallel mutations, even if rare, were more frequent than expected.

**Genetic diversification of naturally evolved *Mimosa* symbionts.** We sequenced, or collected from public databanks, the genomes of 58 *Cupriavidus* strains to study the genetic changes associated with the natural emergence of *Mimosa* symbionts in the genus and to compare them with those observed in the experiment (see Supplementary Note 1 and associated tables for data sources, coverage, and details of the results). We identified 1844 orthologs present in all genomes of the genus (genus core genome). The phylogeny of the genus, using this core genome, was very well resolved since only a few nodes within *C. taiwanensis* show values of bootstrap lower than 90% (Supplementary Fig. 1 and Supplementary Data 2). The tree shows that 44 out of the 46 genomes with the *nod* and *nif* genes were in the monophyletic *C. taiwanensis* clade (Fig. 2). The two exceptions, strains UYPR2.512 and amp6, were placed afar from this clade in the phylogenetic tree (Fig. 2 and Supplementary Data 3). Several of them were shown to be bona fide symbionts since they fix nitrogen in symbiosis with their host[27,28].

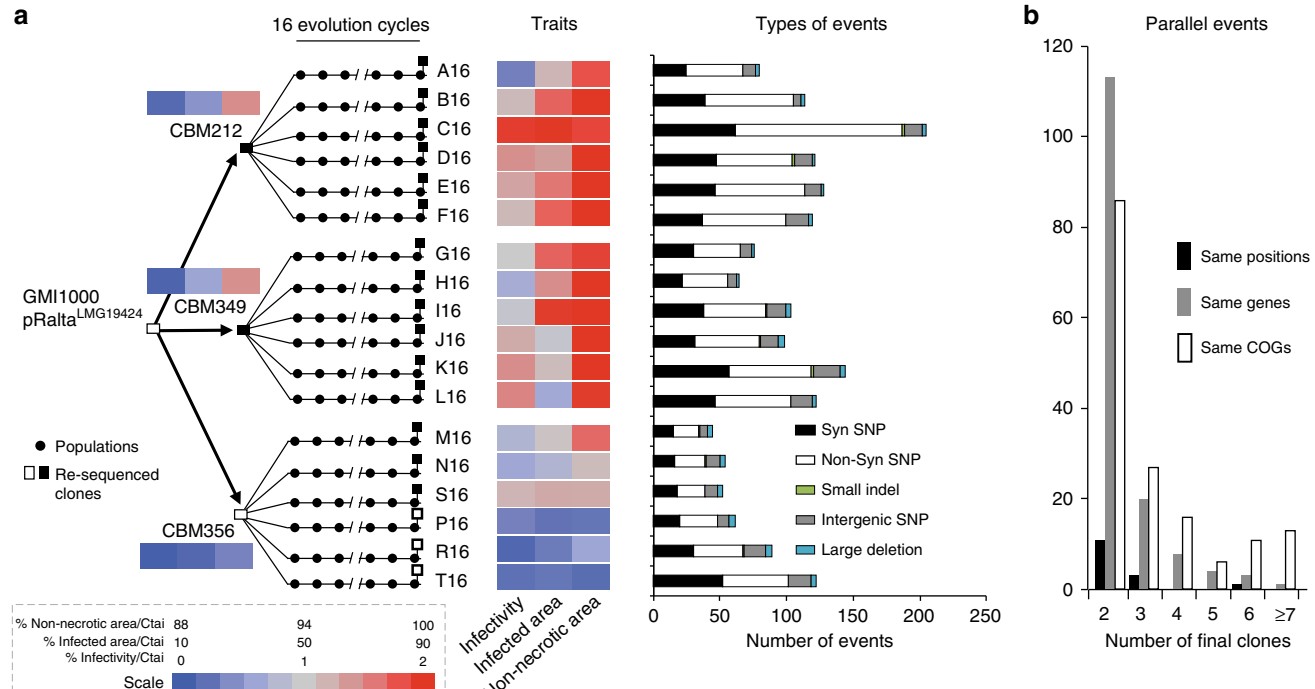

**Fig. 1** Experimental evolution of *Ralstonia* and associated symbiotic and genomic changes. **a** An ancestral chimeric clone evolved to give origin to three clones able to nodulate *M. pudica*. These clones were then evolved in 18 independent lineages using 16 serial nodulation cycles. This process led to improved infectivity (number of viable bacteria recovered per nodule) and intracellularly infected area per nodule section and a decrease of necrotic area per nodule section (heatmap on traits). Except clones CBM356, P16, R16 and T16 (white squares), all acquired the ability of intracellular infection (black squares). The events identified at the end of the 16 evolution cycles for each lineage are indicated on the right (see list of deletions in Supplementary Table 1 and other mutations in Supplementary Data 1). **b** Black bars indicate the number of nucleotides mutated at the same position in two or more clones. Gray bars indicate the number of genes mutated in two or more clones. White bars indicate the number of COG functions with mutations in two or more clones

To prepare the comparisons between experimental and natural evolution we characterized the levels of genetic diversity within *C. taiwanensis* strains. Unexpectedly, the average nucleotide identity (ANIb) values between *C. taiwanensis* strains were often lower than 94%, showing the existence of abundant polymorphism and suggesting that *C. taiwanensis* is not a single species, but a complex of several closely related ones (Fig. 2, Supplementary Figs. 2 and 3, Supplementary Note 1, Supplementary Data 4). We then identified the genes in the core genome (those with orthologs in all strains), and in the pan genome (those present in at least one strain) of *C. taiwanensis*. Together, *C. taiwanensis* strains had a core genome of 3568 protein families and a large pan genome, 3.4 times larger than the average genome. Hence, this complex of species has very diverse gene repertoires and core genes that accumulated more genetic diversity than would be expected for a single bacterial species.

It was proposed that *C. taiwanensis* evolved as a symbiont recently, following the acquisition of the symbiotic genes[19,29]. To test this hypothesis, we first evaluated how many times the rhizobial character (defined by the presence of the core genes of the symbiotic locus *nod-nif-fix*) was independently acquired in the genus *Cupriavidus* (Fig. 2 and Supplementary Fig. 1). This analysis involves phylogenetic reconstruction of the genus and inference of ancestral states (presence of rhizobial genes). The phylogenetic inference is very robust; the three clades are separated in the tree by nodes of high statistical confidence (100% bootstrap, Supplementary Fig. 2). To reconstruct ancestral states, we used birth−death models that describe the rate of gain and loss of genes in a tree using maximum likelihood (see Methods). This analysis showed that the most likely

reconstruction of the character in the phylogenetic tree involves three independent transitions towards symbiosis in the branch connecting the last common ancestor of *C. taiwanensis* and its immediate ancestor (branch before LCA$^{Ct}$, hereafter named bLCA$^{Ct}$), and in the terminal branches leading to strains UYPR2.512 and amp6 (Supplementary Fig. 4). In agreement with this proposition, we found very few homologs of the 514 pRalta$^{LMG19424}$ genes in the genomes of UYPR2.512 (8.3 %) or amp6 (6.4%) once the 32 symbiotic genes were excluded from the analysis. Furthermore, the sequence similarity between these few homologs was significantly smaller than those of core genes ($P <$ 0.01, Wilcoxon test).

We then used the birth−death models to identify all acquisitions of genes in the branch bLCA$^{Ct}$ (Fig. 2 and Supplementary Data 5). This analysis highlighted a set of 435 gene acquisitions that were present in pRalta$^{LMG19424}$, over-representing functions such as symbiosis, plasmid biology, and components of type IV secretion systems (Supplementary Data 6). These results are consistent with a single initial acquisition of the plasmid in this clade at the branch bLCA$^{Ct}$. PacBio resequencing of five strains representative of the main lineages, one for each putative novel species of *C. taiwanensis*, confirmed the ubiquitous presence of a variant of pRalta encoding the symbiotic genes (Supplementary Table 2). Finally, while most individual *C. taiwanensis* core gene trees showed some level of incongruence with the concatenate core genome tree (2897 out of 3568, SH test), an indication of recombination, this frequency was actually lower in the core genes of the plasmid (SH, $(O − E)/(O + E) = −0.41$, significant difference: $P < 0.04$, Fisher's exact test). Similarly, there were signals of

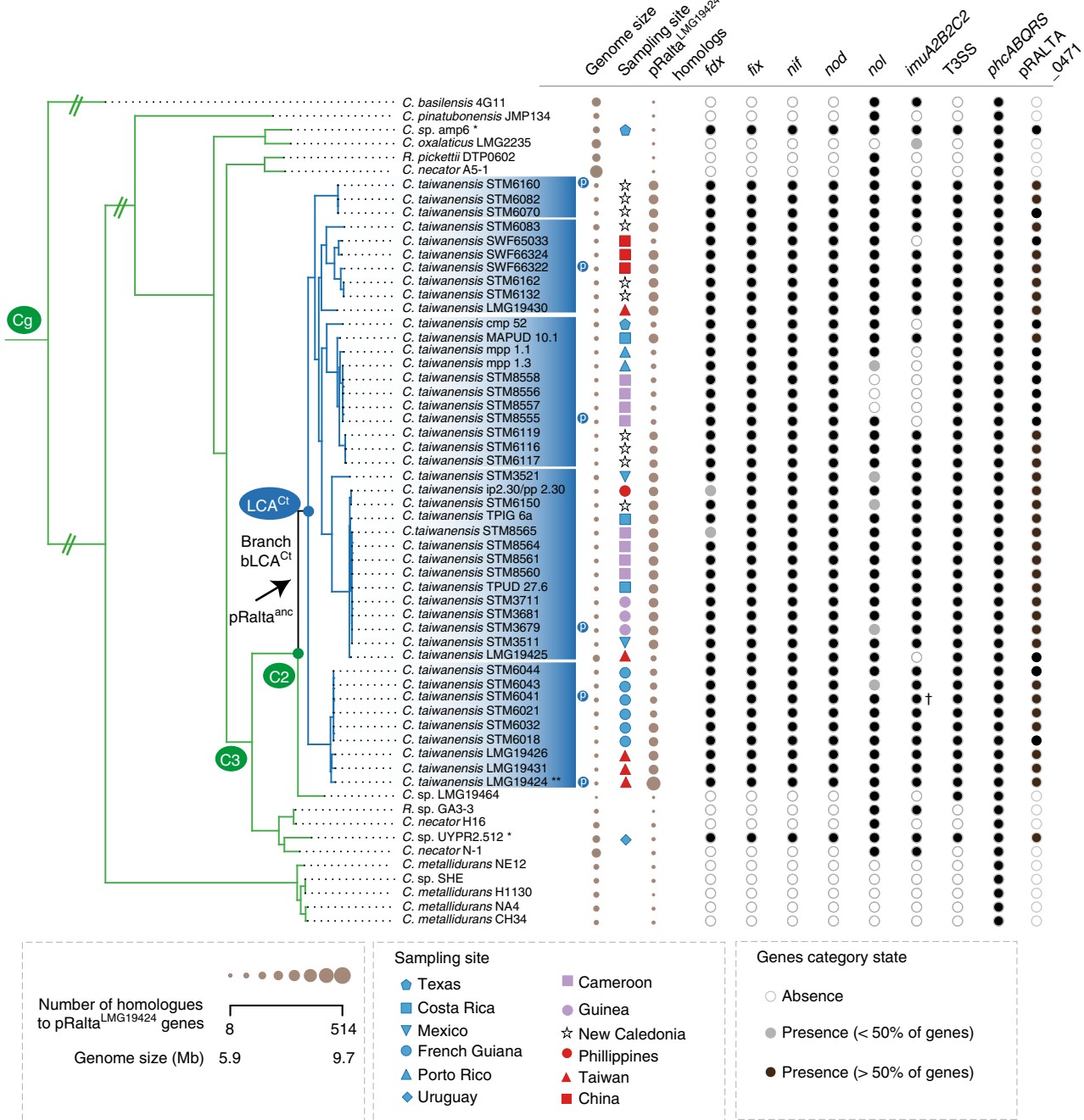

**Fig. 2** Distribution of symbiotic genes, the mutagenic *imuA2B2C2* cassette, T3SS, and *phcABQRS* within the 60 strains of *Cupriavidus*. See Supplementary Fig. 2 for the complete tree of the genus *Cupriavidus* and *Ralstonia* without simplifications in branch length. The arrow indicates the most parsimonious scenario for the acquisition of the pRalta ancestor (inferred using birth-death models of symbiotic gene acquisitions and the MPR function of the ape package in R). This is the branch before the LCA^Ct. The node LCA^Ct indicates the last common ancestor of *C. taiwanensis*. Circles indicate absence (white), presence of less than 50% of the genes (light gray) and presence of more than 50% of the genes (black). Note that most rhizobia possess the *pRalta_0471* gene which is located downstream a *nod* box in LMG19424. The size of the circles for Genome size and pRalta homologs is proportional to the value of the variable. Sampling sites are coded according to geographic origins. Clusters were computed according to different thresholds of ANIb (as indicated in the Results and in Supplementary Figs. 2 and 3). Symbols: Ct, C2, C3, and Cg: LCA of clades analyzed in this study. p (in a blue circle): plasmid re-sequenced by PacBio. *: two rhizobia are not part of *C. taiwanensis*. **: *C. taiwanensis* type strain used as pivot to compute searches of orthologs. † In the PacBio version of this genome *imuA2B2C2* is very similar to that of the reference strain, but is encoded in another plasmid

intragenic recombination in many genes of the core genome (1699 out of 3568, pairwise homoplasy index (PHI) test), but these were significantly less frequent in plasmid core genes (PHI, $(O - E)/(O + E) = -0.92$, significant difference: $P < 0.001$, Fisher's exact test). This suggests that the plasmid inheritance was mostly vertical within *C. taiwanensis*.

Particularly, the observation that plasmid core genes are more congruent with the core genome tree than the other core genes further reinforces the scenario of a single ancient integration of a pRalta ancestor at the LCA^Ct. We thus concluded that the three rhizobial clades evolved independently and that the acquisition of the ancestral symbiotic plasmid of *C. taiwanensis*

should be placed at the branch bLCA[Ct]. The date of plasmid acquisition was estimated using a 16S rRNA genes clock in the range 12–16 MY ago. Although these dating procedures are only approximate, the values are consistent with the low ANIb values within *C. taiwanensis*. They place the acquisition of the plasmid after the radiation of its most typical host (*Mimosa*[30]).

**Parallel evolution after acquisition of the symbiotic plasmid.** Since the experiment only reproduced the initial stages of symbiogenesis, we put forward the hypothesis that the parallels between experimental and natural adaptation should be most striking at the branch bLCA[Ct], i.e., during the onset of natural evolution towards symbiosis. In the experiment, a mechanism of transient hypermutagenesis was shown to accelerate the symbiotic evolution of bacterial populations under plant selective pressure[24]. The symbiotic plasmid transferred to *R. solanacearum* carries a *imuA2B2C2* cassette encoding stress-responsive error-prone DNA polymerases that increased the mutation rate of the recipient genome in the rhizosphere. The long time span since the acquisition of the plasmid in nature and the lack of internal time-calibration points precluded the analysis of accelerated evolution in the branch bLCA[Ct] (relative to others). However, we could identify the *imuA2B2C2* cassette in most extant strains. To evaluate the possibility that symbiotic plasmids with and without the mutagenesis cassette were independently acquired, we analyzed the patterns of sequence similarity between genes in the *imuA2B2C2* cassette and in *nif-nod* locus and compared these values with the distribution of similarity in all core genes. The results show distribution of sequence similarity in these genes as expected if they were acquired only once (they are within the range of variation of core genes, Supplementary Figs. 5 and 6). Analysis of the genomes re-sequenced using PacBio technology confirmed the presence of the *imuA2B2C2* locus on the symbiotic plasmid, except in strain STM6041, where this cassette was present on another plasmid (and showed lower sequence similarity, Supplementary Fig. 5). The conservation of the plasmid cassette suggests it has been under selection. Whether it has played a role in the symbiotic evolution of *Cupriavidus* cannot be assessed in this study, because we lack calibration points in the tree to infer an acceleration in the rate of molecular evolution in the branch bLCA[Ct].

We put forward the hypothesis that genes with an excess of polymorphism in the branch bLCA[Ct] (relative to the other branches) were more likely to have endured adaptive changes. To identify such genes, we took all the genes in the core genome that were present in the branch bLCA[Ct] (core genome C3 in Supplementary Data 2) and computed their genetic diversity in *C. taiwanensis*. We also computed the ancestral sequences of these genes, accounting for recombination using ClonalFrameML, at the node LCA[Ct] and at the node immediately ancestral (C2 in Fig. 2). We then computed for each gene the number of changes between these two nodes (these are the changes that accumulated in the branch bLCA[Ct]), and compared this value to the diversity of genes in *C. taiwanensis*. This analysis revealed 67 genes with a clear excess of changes in bLCA[Ct] relative to the expected values given the diversity in the species (Supplementary Fig. 7 and Supplementary Data 5). They corresponded to genes with an excess of polymorphism in bLCA[Ct]. To study the parallels between the experimental and natural processes, we identified the 2372 orthologs between *R. solanacearum* and *C. taiwanensis* (Supplementary Data 7), and added the 514 pRalta genes in the chimera as orthologs. These are the genes that have orthologs in the two

systems and can thus be queried to identify parallels. We found that final clones of the evolution experiment had significantly more mutations in genes whose orthologs had an excess of polymorphism at the onset of symbiosis in natural populations (P < 0.001, Fisher's test; Supplementary Tables 3 and 4). Hence, there is a significant overlap in the genes that mutated in the experiment and diverged quickly in nature upon plasmid acquisition. This revealed a first parallel between the natural and experimental processes.

The genomic rates of non-synonymous substitutions in natural populations are systematically smaller than those of synonymous substitutions[31,32]. Accordingly, the substitutions in the core genes of *C. taiwanensis* showed an excess of synonymous changes (Supplementary Fig. 8). In contrast, experimental evolution studies often show that adaptation occurs by the fixation of an excess of non-synonymous changes[25,33–36], including in *R. solanacearum*[37]. Yet, we identified an excess of synonymous mutations over non-synonymous mutations in the evolution experiment[20]. Both processes are thus characterized by a predominance of purifying selection.

**Adaptation occurred in the genetic background, not in the symbiotic plasmid.** The symbiotic plasmids carry many genes and induce a profound change in the lifestyle of the bacteria. We thus expected to identify changes in the plasmid reflecting its accommodation to the novel genetic background. The plasmid pRalta[LMG19424] accumulated an excess of synonymous substitutions and the majority of the genetic deletions observed in the experiment (Fig. 3 and Supplementary Table 5). Interestingly, whereas core genes in the *nod-nif* locus were very conserved in natural isolates (Supplementary Fig. 6), this locus had many mutations in the experiment (Supplementary Data 1). Natural populations also showed more deletions in the plasmid, since from the 413 genes present in pRalta[LMG19424] and inferred to be present in LCA[Ct], only 12% were in the core genome. This is six times less than found among the chromosomal genes present in *C. taiwanensis* LMG19424 and inferred to be present in LCA[Ct] (P < 0.001, Fisher's exact test, Fig. 3b). Notably, the few pRalta[LMG19424] core genes are related to the symbiosis or to typical plasmid functions (conjugation) (Fig. 3).

In order to test if the observed rapid plasmid genetic diversification was driving the adaptation to symbiosis in natura, we compared the rates of positive selection on plasmid and chromosomal genes in *C. taiwanensis*. We identified 325 genes under positive selection in the clade, and 46 specifically in the branch bLCA[Ct] (analysis of 1869 and 1676 core genes lacking evidence of recombination using PHI, respectively, Supplementary Data 5). Surprisingly, all 325 genes under positive selection were chromosomal (none was found among the core genes of the plasmid, P = 0.001, $\chi^2$ test, Fig. 3e). This suggests that adaptation took place mostly in the chromosomes.

To test this hypothesis in the experimental study, we re-analyzed all mutations previously identified as adaptive in the evolution experiment. They were all chromosomal[18,22,23]. Since our previous analyses of mutations identified in the evolution experiment only focused on strongly adaptive genes, we evaluated the impact of pRalta[LMG19424] mutations on the symbiotic evolution of *R. solanacearum* by replacing the evolved plasmid with the original pRalta[LMG19424] in three evolved clones (B16, G16, and I16, thus generating strains B16-op, G16-op, and I16-op, respectively). The relative in planta fitness of the new chimeras harboring the original plasmid were not significantly different from that of the experimentally evolved clones (Fig. 3f), showing that the adaptation of these strains did not involve mutations in

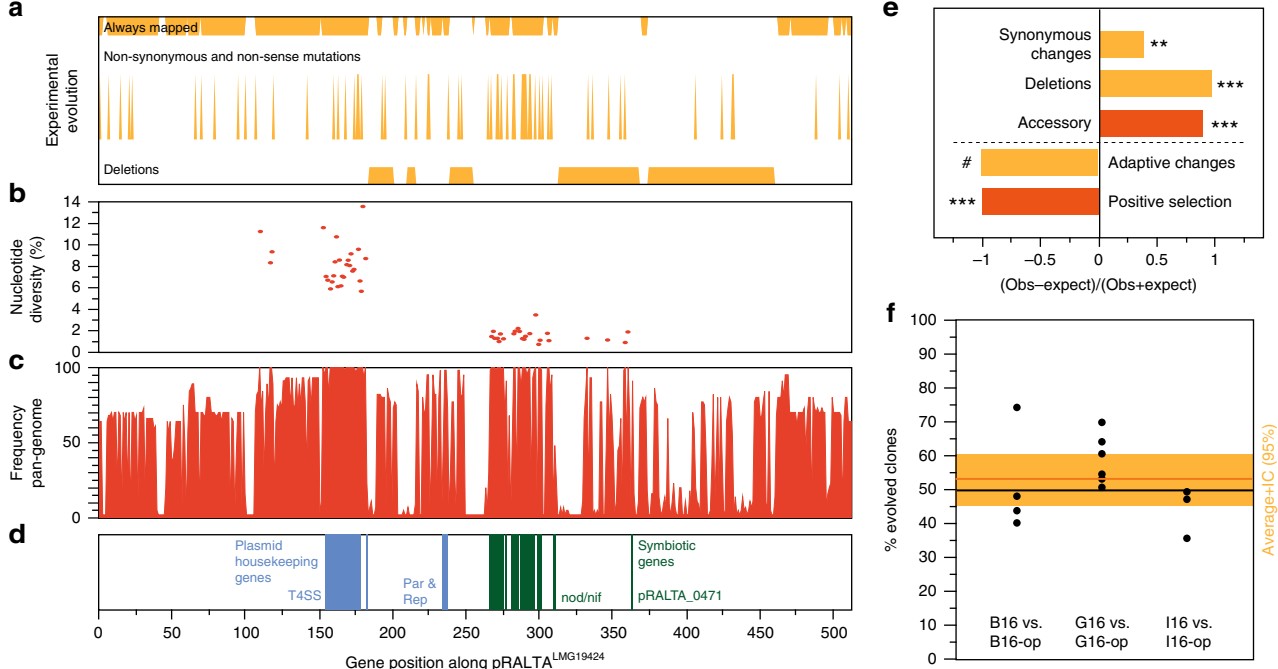

**Fig. 3** Analysis of the symbiotic plasmid of *C. taiwanensis* LMG19424. **a** Deletions, non-synonymous and non-sense mutations, and regions of the plasmid that could always be mapped to identify mutations in the experiment. **b** Nucleotide diversity of natural *C. taiwanensis* core genes: symbiotic genes accumulated much less diversity than the other genes. **c** Frequency of each gene in the 44 *C. taiwanensis* (positional orthologs). **d** Symbiotic and plasmid housekeeping genes. **e** Observed over expected values for a number of traits in the plasmid natural (red) or experimental (orange) evolution (Supplementary Tables 1, 4, and Data 5). **/***significantly different from 1 (*P* < 0.01/0.001, Fisher's exact tests for all but the test for "Synonymous changes" which was made by permutations, see Methods and Supplementary Table 5). # We could not find a single adaptive mutation in the plasmid in our previous works neither in the experiments in panel (**f**). **f** Impact of pRalta mutations on the in planta fitness of evolved clones. *M. pudica* plantlets were co-inoculated with pairs of strains at a 1:1 ratio and nodules were harvested at 21 dpi for bacteria counting. Each pair consisted of an evolved clone (B16, G16 or I16) and the same clone with the evolved pRalta replaced by the original one (B16-op, G16-op or I16-op). Each dot represents an independent experiment. The orange horizontal bar represents the average and the large orange rectangle the 95% interval of confidence of the average (*t* Student). This interval includes the value 50% indicating that the two types of clones are not significantly different in terms of fitness

the plasmid. Importantly, the original chimera had similar ex planta survival rates with and without the plasmid (Supplementary Fig. 9), showing that plasmid carriage does not have a fitness cost in this respect (Supplementary Tables 6 and 7). Although we cannot exclude that some events of positive selection in the plasmid may have passed undetected, nor that further symbiotic evolution of *Ralstonia* chimera will involve plasmid mutations, it appears that the genetic changes leading to improvement of the symbiotic traits occurred mainly in the chromosomes of the chimera in the experiment, and of *C. taiwanensis* in nature, not on the plasmid carrying the symbiotic traits.

**Parallel co-option of regulatory circuits**. To search for parallel adaptive mutations, we first analyzed the 436 genes with non-synonymous or non-sense mutations in the experiment (Supplementary Data 1). This set of genes over-represented virulence factors of *R. solanacearum* (47 genes), including the T3SS effectors, EPS production, and a set of genes regulating (*phcBQS*) or directly regulated (*prhI*, *hrpG*, and *xpsR*) by the central regulator PhcA of the cell density system that controls virulence and pathogenicity in *R. solanacearum*[38] (Fig. 4a and Supplementary Data 8). Among this set of genes, mutations in the structural T3SS component *hrcV*, or affecting the virulence regulators *hrpG*, *prhI*, *vsrA*, and *efpR*, were demonstrated to be responsible for the acquisition or the drastic improvement of nodulation and/or infection[18,22,23]. Only 10 of the 47 mutated

virulence-associated genes have an ortholog in *C. taiwanensis*: *vsrD*, *xpsR*, Rsp0736, *tssD*, three *phc* genes (*phcQ*, *phcB*, and *phcS*) and three structural genes of the T3SS (*hrcV*, *hrcR*, and *hrpQ*). We focused on the *phc* system and the T3SS to evaluate their respective roles in the experimental and natural processes.

Silencing of the T3SS and its effectors was required to activate symbiosis in the evolution experiment, presumably because some effectors block nodulation and early infection[18]. We searched for the T3SS in the *Cupriavidus* genomes to test if the onset of symbiosis was associated with the acquisition or the loss of a T3SS. In contrast to the evolution experiment, the emergence of legume symbiosis in natura seems to be associated with the acquisition of T3SS since all rhizobial *Cupriavidus* strains of our sample encode a (chromosomal) T3SS, while most of the other *Cupriavidus* strains do not (Fig. 2). To understand this difference between the two processes, we searched for orthologs of the 77 T3SS effectors of *R. solanacearum* GMI1000 in *C. taiwanensis* LMG19424, but we found no single ortholog for these genes (Supplementary Data 5). In complement, it has been shown that a functional T3SS is not required for mutualistic symbiosis of the latter with *M. pudica*[39], the only plant species used in the evolution experiment. Hence, the differences between the two processes seem to be caused by selection for silencing some *R. solanacearum* effectors that are lacking in *C. taiwanensis*.

We then focused on PhcA-associated genes since they accumulated an excess of mutations in the experiment (Supplementary Data 8). The *phc* system, which was only found intact in

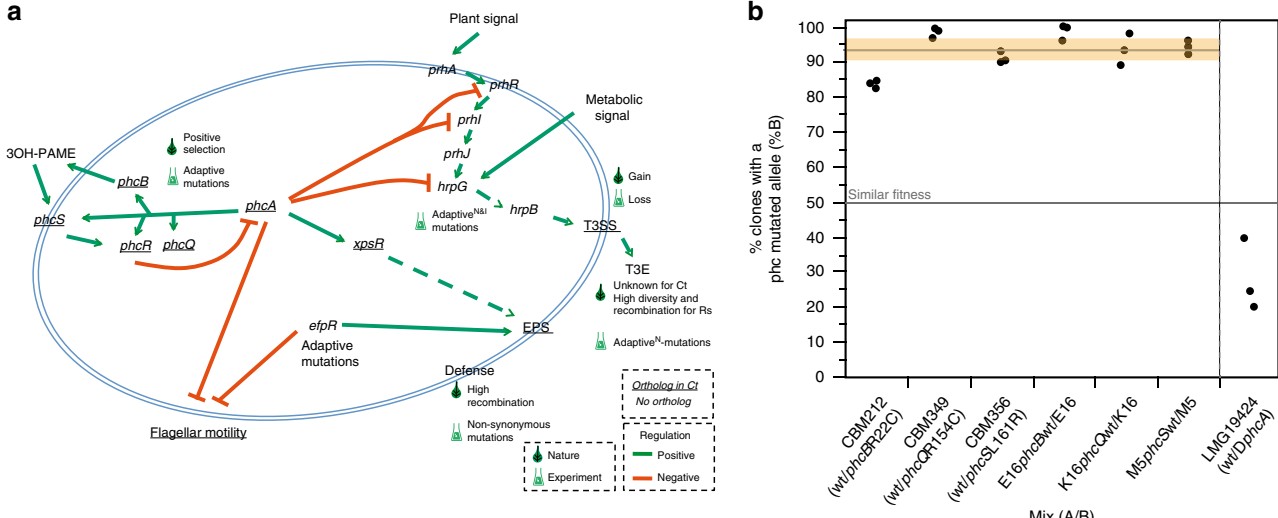

**Fig. 4** Virulence factors and regulatory pathways of *R. solanacearum* and their evolution in the evolution experiment. **a** Schema of the major virulence factors and regulatory pathways mentioned in this study and their role in *R. solanacearum* (adapted from ref. [38]). Adaptive[N] and adaptive[I] represent the presence of adaptive mutations for nodulation (N) and infection (I), respectively. Underline, genes or factors present in *C. taiwanensis*. The results of the enrichment analyses are in Supplementary Data 6, 8, and 9. **b** Adaptive nature of the *Ralstonia phc* alleles evolved in the experiment and the recruitment of PhcA for symbiosis in the natural symbiont *C. taiwanensis* LMG19424. Each dot represents an independent experiment. The horizontal gray line represents the average fitness of the evolved *phc* genes relative to the wild type. The horizontal orange rectangle indicates the 95% interval of confidence for the mean. The results for *phc* are significantly different from the expected under the hypothesis that both variants are equally fit (horizontal line at 50%, $P < 0.005$, Wilcoxon test). The mean for the analysis of the mutant of PhcA (25%) is smaller than 50%, although the difference is at the edge of statistical significance ($P = 0.0597$, two-side *t* Student's test). The codes of the clones correspond to those indicated in Fig. 1

*Cupriavidus* and *Ralstonia* (Supplementary Data 10), regulates a reversible switch between two different physiological states via the repression of the central regulator PhcA in *Ralstonia*[38] and *Cupriavidus*[40]. Interestingly, PhcA-associated genes were also enriched in substitutions in natura. Indeed, the *phcBQRS* genes of the cell density-sensing system were among the 67 genes that exhibited an excess of nucleotide diversity in the branch bLCA[Ct] relative to *C. taiwanensis* ("phcA-linked" in Supplementary Data 6). Strikingly, only seven genes showing an excess of diversity at bLCA[Ct] had orthologs with mutations in the evolution experiment. Among these seven, only two also showed a signature of positive selection in *C. taiwanensis*: *phcB* and *phcS* (ongoing events, Supplementary Data 5).

Given the parallels between experimental and natural evolution regarding an over-representation of changes in PhcA-associated genes, we enquired on the possibility that mutations in the *phcB*, *phcQ* and *phcS* genes, detected in the evolved E16, K16 and M16 clones capable of nodule cell infection were adaptive for symbiosis with *M. pudica*. For this, we introduced the mutated alleles of these genes in their respective nodulating ancestors, CBM212, CBM349, and CBM356, and the wild-type allele in the evolved clones E16, K16, and M5 (M5 was used instead of M16, since genetic transformation failed in the latter clone in spite of many trials). Competition experiments between the pairs of clones harboring the wild-type or the mutant alleles confirmed that these mutations were adaptive (Fig. 4b). The evolved clones also showed better infectivity as measured by the number of bacteria per nodule (Supplementary Fig. 10). On the other hand, we found that the Phc system plays a role in the natural *C. taiwanensis*-*M. pudica* symbiosis, since a *phcA* deletion mutant had lower in planta fitness than the wild-type *C. taiwanensis* (Fig. 4b), and lower infectiveness (Supplementary Fig. 11), when both strains were co-inoculated to *M. pudica*. Hence, the rewiring of the *phc* virulence regulatory pathway of *R. solanacearum* was involved in the evolution of symbiosis in several lineages of the

experimental evolution. In parallel, high genetic diversification accompanied by positive selection of the homologous pathway was associated with the transition to symbiosis in the natural evolution of *C. taiwanensis*.

## Discussion

Years of comparative genomics and reverse genetics approaches led to propose that most legume symbionts evolved in two-steps[12], i.e., acquisition of a set of essential symbiotic genes by horizontal transfer followed by subsequent adaptation of the resulting genome under plant selective pressure. Recent experimental work was able to confirm this scenario up to the point where plants nodulate and bacteria produce intracellular infection[18,22]. Yet, it was unclear if there were parallels between the experimental and the natural evolution. Such parallels were not necessarily expected, because the two processes differed in a number of fundamental points. The two species are from different genera, share only 2140 orthologues (excluding pRALTA), and had different original lifestyles, saprophytic for *C. taiwanensis* and pathogenic for *R. solanacearum*. The conditions of the experimental evolution were extremely simplified and controlled, whereas natural environmental conditions were certainly very complex and changing. The time span of both processes was radically different, 12–16 MYA in nature, and ca. 400 bacterial generations per lineage in the experiment, providing very different magnitudes of genetic diversity. *C. taiwanensis* are well-adapted mutualistic symbionts of *Mimosa* spp., whereas the lab-evolution of *Ralstonia* is not yet achieved, none of the evolved clones being able to persist within nodule cells and fix nitrogen to the benefit of the plant. In spite of these differences, we highlighted several parallels between the experimental and in natura transitions towards legume symbiosis (Fig. 5). We also highlighted some clear differences—concerning the T3SS and its effectors—and were unable to test some parallels because of the

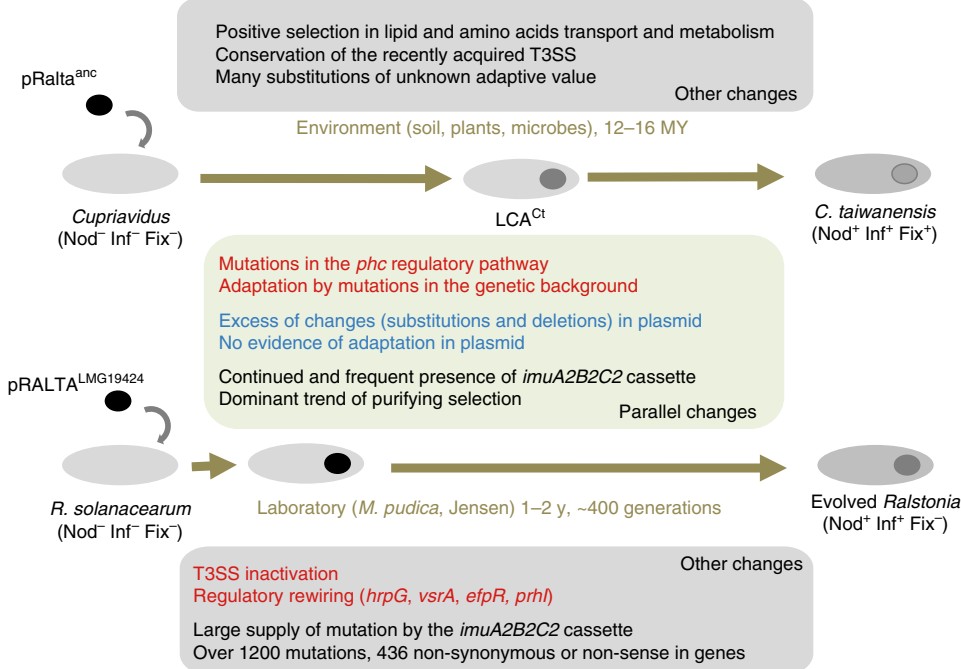

**Fig. 5** Overall similarities and differences between the experimental and natural evolutionary processes described in this study. Known adaptive and non-adaptive changes are in red and blue, respectively

high genetic diversity identified in natural populations, e.g., high mutation rates at bLCA$^{Ct}$ possibly caused by the *imuA2B2C2* cassette encoding stress-responsive error-prone DNA polymerases. We focus our discussion on the parallels between natural and experimental processes because there is little published data regarding them.

The plasmid carrying the essential *nod* and *nif* genes drove the transition towards symbiosis in both processes. We expected that plasmid genes would show evidence of adaptation to the novel genetic background and environmental conditions. Instead, the abundant substitutions observed in the plasmid seem to have a negligible role in the experimental symbiotic adaptation and lack evidence of positive selection in nature. Moreover, the plasmid did not seem to have a cost in ex planta culture conditions. This suggests that the symbiotic genes acquired by *C. taiwanensis* in nature were already—like in the experiment—pre-adapted to establish a symbiotic association with *Mimosa* species. This is in agreement with proposals that pRalta was acquired from *Burkholderia*[41], which are ancient symbionts of *Mimosa* spp.[42]. This also suggests that adaptation following the acquisition of a large plasmid encoding traits driving ecological shifts does not require plasmid evolution. The fact that genetic adaptation to this novel complex trait only occurred in the background is a testimony of the ability of mobile genetic elements to seamlessly plug novel functions in their hosts.

Adaptive mutations were found on regulatory modules, the rewiring of which may inactivate or co-opt native functions for the novel trait. We previously showed that loss of the ability to express the T3SS was strictly necessary for the early transition towards symbiosis in the experiment[18], and that subsequent adaptation favored the re-use of regulatory modules leading to massive metabolic and transcriptomic changes[22]. These phenotypic shifts occurred via mutations targeting regulatory genes specific to *Ralstonia* (e.g., *hrpG*, *prhI*, *efpR*, Rsc0965), which finely control the expression of many virulence determinants[38,43,44]. Here, from the analysis of orthologs between *Ralstonia* chimera and *C. taiwanensis*, we showed that several genes in the *phcBQRS*

operon both exhibited significant positive selection in *C. taiwanensis* populations and accumulated adaptive mutations in the evolution experiment. In *R. solanacearum*, these genes control the activity of the global virulence regulator PhcA via a cell density-dependent mechanism[45]. Mutations in this pathway are unlikely to cause adaptation by attenuating the virulence of *Ralstonia*, because the chimeric ancestor is not pathogenic on *M. pudica*[18]. Since PhcA also plays a role in the natural *C. taiwanensis—M. pudica* symbiosis, we speculate that adaptive mutations in the experiment and high genetic diversification in nature on *phc* genes after the acquisition of pRalta may reflect the rewiring of a quorum-sensing system to sense the environment for cues of when to express the novel mutualistic dialogue with eukaryotes. Further work should determine if some of these mutations resulted in the integration of the gene expression network of the plasmid in the broader network of the cell.

Very controlled experimental evolution studies show few identical mutations between replicates and require the aggregation of mutations in genes, operons or pathways to identify commonalities[25]. This was the case of the initial evolution of three clones nodulating *Mimosa*, which occurred by three mutations in two different genes of the virulence pathway of *R. solanacearum*[18]. Here, the comparison of the natural evolution of *Mimosa* symbionts in the *Cupriavidus* genus and the experimental symbiotic evolution of *Ralstonia* under *M. pudica* selection pressure could not reveal parallel changes at the nucleotide level because of the high diversity of natural populations. Yet, it showed that symbiotic adaptation occurred in the recipient genome, in a selection regime where most non-synonymous mutations were purged (purifying selection) but some were adaptive and allowed the continuous improvement of the symbiotic traits in the experiment and, presumably given its efficiency, in nature. Adaptive changes affected homologous genes implicated in central regulatory pathways in both processes highlighting how natural selection can lead to the co-option of homologous structures in different lineages submitted to similar evolutionary challenges. The observation of these parallels

highlights the potential of research projects integrating population genomics, molecular genetics, and evolution experiment to provide insights on adaptation in nature and in the laboratory. Therefore, experimental evolution appears not only useful to demonstrate the biological plausibility of theoretical models in evolutionary biology, but also to enlighten the natural history of complex adaptation processes.

## Methods

**Dataset for the experimental evolution.** We used previously published data on the genomic changes observed in the experimental evolution of the chimera, including 21 bacterial clones (three ancestors and 18 evolved clones)[20]. We analyzed all the synonymous and non-synonymous mutations of each clone from these datasets (Supplementary Data 1). Large deletions above 1 kb were first listed based on the absence of Illumina reads in these regions, and were then validated by PCR amplification using specific primers listed in Supplementary Table 8. Primers were designed to amplify either one or several small fragments of the putative deleted regions or the junction of these deletions. All primer pairs were tested on all ancestors and final clones (Supplementary Table 1).

**Mutant construction.** Strains used in this study are indicated in Supplementary Table 6. The pRalta in evolved *Ralstonia* clones B16, G16, and I16 was replaced by the wild-type pRalta of *C. taiwanensis* LMG19424 strain as previously described[20], generating B16-op, G16-op, and I16-op. Wild-type alleles of the *phcB*, *phcQ*, and *phcS* genes and constitutively expressed reporter genes (GFP, mCherry) were introduced into *Ralstonia* evolved clones using the MuGent technique[46]. Briefly, this technique consisted in the co-transformation of two DNA fragments, one fragment carrying a kanamycin resistance cassette together with a gene coding a fluorophore and one unlabeled PCR fragment of ca. 6 kb carrying the point mutation to introduce, as previously described[22]. Co-transformants were first selected on kanamycin, then screened by PCR for the presence of the point mutation. M5, which possesses the *phcS* mutation, was used instead of M16 since M16 is no more transformable.

To construct the *phcA* deletion mutant of LMG19424, we used the pGPI-SceI/pDAI-SceI technique[47]. Briefly the regions upstream and downstream *phcA* were amplified with the oCBM3413-3414 and oCBM3415-3416 primer pairs and the Phusion DNA polymerase (Thermo Fisher Scientific). The two PCR products were digested with *Xba*I-*Bam*HI and *Bam*HI-*Eco*RI respectively and cloned into the pGPI-SceI plasmid digested by *Xba*I and *Eco*RI. The resulting plasmid was introduced into LMG19424 by triparental mating using the pRK2013 as helper plasmid. Deletion mutants were obtained after introduction of the pDAI-SceI plasmid encoding the I-SceI nuclease. LMG19424 *phcA* deletion mutants were verified by PCR using the oCBM3417-3418 and oCBM3419-3420 primer pairs corresponding to external and internal regions of *phcA*, respectively. Oligonucleotides used in these constructions are listed in Supplementary Table 9.

**Relative in planta fitness.** *Mimosa pudica* seeds from Australia origin (B&T World Seed, Paguignan, France) were surface sterilized for 15 min in concentrated sulfuric acid, rinsed with sterile water and incubated for further 10 min in 2.4% sodium hypochlorite solution. Seeds were soaked for 24 h in sterile water at 28 °C under agitation then germinated on soft agar plates in darkness at 28 °C for 24 h. Seedlings were grown in N-free conditions in Gibson tubes filled with 20 ml of Fahraeus solid medium[48] and 50 ml of 1/4th strength of Jensen liquid medium[49]. To measure the in planta relative fitness, a mix of two strains bearing different antibiotic resistance genes or fluorophores ($5 \times 10^5$ bacteria of each strain per plant) were inoculated to 20 plants. All nodules were harvested 21 days after inoculation, pooled, surface sterilized, and crushed. Dilutions of nodule crushes were spread on selective plates, incubated 2 days at 28 °C, then colonies were counted using a fluorescent stereo zoom microscope V16 (Zeiss) when needed. Three independent experiments were performed for each competition.

**Survival measurement.** Single colonies of GMI1000 (RCM1068) or GMI1000-pRalta (RCM1069) were grown overnight in rich BG medium[10] and $10^7$ bacteria were inoculated to Gibson tubes filled with quarter-strength Jensen medium[24] alone or containing in addition two *M. pudica* plantlets. Bacteria were counted by plating. Twelve independent experiments were performed.

**Public genome dataset.** We collected 13 genomes of *Cupriavidus* spp. (including three rhizobia) and 31 of *Ralstonia* from GenBank RefSeq and the MicroScope platform (http://www.genoscope.cns.fr/agc/microscope/home/) as available in September 2015. We removed the genomes that seemed incomplete or of poor quality, notably those smaller than 5 Mb and with L90 > 150 (defined as the smallest number of contigs whose cumulated length accounts for 90% of the genome). All accession numbers are given in Supplementary Data 3. Genomes of α- and β-Proteobacteria larger than 1 Mb and genomes of phages were downloaded from GenBank RefSeq as available in February 2013.

**Production and analysis of Illumina sequences.** The genomes of 43 *Mimosa* spp. isolates, a non-rhizobial strain of *Cupriavidus* (strain LMG19464) as well as a *C. oxalaticus* strain (LMG2235) (Supplementary Data 3), were sequenced at the GeT-PlaGe core facility, INRA Toulouse (get.genotoul.fr). DNA-seq libraries were prepared according to Bioscientific's protocol using the Bioscientific PCR-free Library Prep Kit. Briefly, DNA was fragmented by sonication, size selection was performed using CLEANNA CleanPCR beads and adaptators were ligated to be sequenced. Library quality was assessed using an Advanced Analytical Fragment Analyser and libraries were quantified by qPCR using the Kapa Library Quantification Kit. DNA-seq experiments were performed on an Illumina HiSeq2000 sequencer using a paired-end read length of $2 \times 100$ bp with the HiSeq v3 reagent kit (LMG2235 and LMG19431) or on an Illumina MiSeq sequencer using a paired-end read length of $2 \times 300$ bp with the Illumina MiSeq v3 reagent kit (other strains). On average, genomes contained 99 contigs and an L90 of 29.

Genome assemblies were performed with the AMALGAM assembly pipeline (Automated MicrobiAL Genome AsseMbler (https://zenodo.org/record/1239599#.WumbmtaxWqq))[50]. The pipeline is a python script (v2.7.x and onward) that launches the various parts of the analysis and checks that all tasks are completed without error. To date AMALGAM embeds SPAdes, ABySS[51], IDBA-UD[52], Canu[53], and Newbler[54]. After the assembly step, an attempt to fill scaffolds/contigs gaps is performed using the gapcloser software from the SOAPdenovo2 package[55]. Only one gap filling round was performed since launching gapcloser iteratively may lead to an over-correction of the final assembly. AMALGAM ends with the generation of a scaffolds/contigs file (fasta format) and a file describing the assembly in agp format (v2.0).

The genomes were subsequently processed by the MicroScope pipeline for complete structural and functional annotation[56]. Gene prediction was performed using the AMIGene software[57] and the microbial gene finding program Prodigal[58] known for its capability to locate the translation initiation site with great accuracy. The RNAmmer[59] and tRNAscan-SE[60] programs were used to predict rRNA and tRNA-encoding genes, respectively.

**PacBio sequencing.** Library preparation and sequencing were performed according to the manufacturer's instructions "Shared protocol-20 kb Template Preparation Using BluePippin Size Selection system (15kb-size cutoff)". At each step, DNA was quantified using the Qubit dsDNA HS Assay Kit (Life Technologies). DNA purity was tested using the nanodrop (Thermo Fisher) and size distribution and degradation assessed using the Fragment analyzer (AATI) High Sensitivity DNA Fragment Analysis Kit. Purification steps were performed using 0.45× AMPure PB beads (Pacbio). Ten micrograms of DNA was purified then sheared at 40 kb using the merarruptor system (diagenode). A DNA and END damage repair step was performed on 5 μg of sample. Then blunt hairpin adapters were ligated to the library. The library was treated with an exonuclease cocktail to digest unligated DNA fragments. A size selection step using a 13–15 kb cutoff was performed on the BluePippin Size Selection system (Sage Science) with the 0.75% agarose cassettes, Marker S1 high Pass 15–20 kb.

Conditioned Sequencing Primer V2 was annealed to the size-selected SMRTbell. The annealed library was then bound to the P6-C4 polymerase using a ratio of polymerase to SMRTbell at 10:1. Then after a magnetic bead-loading step (OCPW), SMRTbell libraries were sequenced on RSII instrument at 0.2 nM with a 360 min movie. One SMRTcell was used for sequencing each library. Sequencing results were validated and provided by the Integrated next generation sequencing storage and processing environment NG6 accessible in the genomic core facility website[61].

**Core genomes.** Core genomes were computed using reciprocal best hits (hereafter named RBH), using end-gap free Needleman−Wunsch global alignment, between the proteome of *C. taiwanensis* LMG19424 or *R. solanacearum* GMI1000 (when the previous is not in the sub-clade) as a pivot (indicated by ** on Supplementary Fig. 2A) and each of the other 88 proteomes[62]. Hits with less than 40% similarity in amino acid sequence or more than a third of difference in protein length were discarded. The lists of orthologs were filtered using positional information. Positional orthologs were defined as RBH adjacent to at least two other pairs of RBH within a neighborhood of ten genes (five up- and five downstream). We made several sets of core genomes (see Supplementary Fig. 2A): all the 89 strains (A1), 44 *C. taiwanensis* (Ct), Ct with the closest outgroup (C2), Ct with the five closest outgroups (C3), the whole 60 genomes of the genus *Cupriavidus* (Cg), and the 14 genomes of *R. solanacearum* (Rs). They were defined as the intersection of the lists of positional orthologs between the relevant pairs of genomes and the pivot (Supplementary Data 2).

**Pan genomes.** Pan genomes describe the full complement of genes in a clade and were computed by clustering homologous proteins in gene families. Putative homologs between pairs of genomes were determined with blastp v2.2.18 (80% coverage), and evalues (if smaller than $10^{-4}$) were used to infer protein families using SiLiX v1.2.8 (http://lbbe.univ-lyon1.fr/SiLiX)[63]. To decrease the number of paralogs in pan genomes, we defined a minimal identity threshold between homologs for each set. For this, we built the distribution of identities for the positional orthologs of core genomes between the pivot and the most distant

genome in the set (Supplementary Fig. 12), and defined an appropriate threshold in order to include nearly all core genes but few paralogs (Supplementary Table 10).

**Alignment and phylogenetic analyses.** Multiple alignments were performed on protein sequences using Muscle v3.8.31 [64], and back-translated to DNA. We analyzed how the concatenated alignment of core genes fitted different models of protein or DNA evolution using IQ-TREE v1.3.8 [65]. The best model was determined using the Bayesian information criterion. Maximum likelihood trees were then computed with IQ-TREE v1.3.8 using the appropriate model, and validated via a ultrafast bootstrap procedure with 1000 replicates[66] (Supplementary Data 2). The maximum likelihood trees of each set of core genes were computed with IQ-TREE v1.3.8 using the best model obtained for the concatenated multiple alignment.

In order to root the phylogeny based on core genes, we first built a tree using 16S rRNA genes of the genomes of *Ralstonia* and *Cupriavidus* genera analyzed in this study and of ten outgroup genomes of β-Proteobacteria. For this, we made a multiple alignment of the 16S rRNA genes with INFERNAL v.1.1 (with default parameter)[67] using RF00177 Rfam model (v.12.1)[68], followed by manual correction with SEAVIEW to remove poorly aligned regions. The tree was computed by maximum likelihood with IQ-TREE using the best model (GTR + I + G4), and validated via an ultrafast bootstrap procedure with 1000 replicates.

To date the acquisition of the symbiotic plasmid in the branch bLCA$^{Ct}$, we computed the distances in the 16S rRNA genes tree between each strain and each of the nodes delimitating the branch bLCA$^{Ct}$ (respectively LCA$^{Ct}$ and C2 in Fig. 2). The substitution rate of 16S rRNA genes in enterobacteria was estimated at ~1% per 50 MY of divergence[69], and we used this value as a reference.

**Orthologs and pseudogenes of specific families of genes.** We identified the positional orthologs of Cg for symbiotic genes, the mutagenic cassette, T3SS, and PhcABQRS using RBH and *C. taiwanensis* LMG19424 as a pivot (such as defined above). These analyses identify bona fide orthologs in most cases (especially within species), and provide a solid basis for phylogenetic analyses. However, they may miss genes that evolve fast, change location following genome rearrangements, or that are affected by sequence assembling (incomplete genes, small contigs without gene context, etc.). They also miss pseudogenes. Hence, we used a complementary approach to analyze in detail the genes of the symbiotic island in the plasmid, the mutagenic cassette, T3SS and PhcABQRS. We searched for homologs of each gene in the reference genome in the other genomes using LAST v744 [70] and a score penalty of 15 for frameshifts. We discarded hits with evalues below $10^{-5}$, with less than 40% similarity in sequence, or aligning less than 50% of the query. In order to remove most paralogs, we plotted values of similarity and patristic distances between the 59 *Cupriavidus* and the reference strain *C. taiwanensis* LMG19424 for each gene. We then manually refined the annotation using this analysis.

**Evolution of gene families using birth−death models.** We used Count (version downloaded in December 2015)[71] to study the past history of transfer, loss and duplication of the protein families of the pan genomes. The analysis was done using the core genomes reference phylogenies. We tested different models of gene content evolution using the tree of Cg (Supplementary Data 2), and selected the best model using the Akaike information criterion (Supplementary Table 10). We computed the posterior probabilities for the state of the gene family repertoire at inner nodes with maximum likelihood and used a probability cutoff of 0.5 to infer the dynamics of gene families, notably presence, gain, loss, reduction, and expansion for the branch leading to the last common ancestor (LCA) of *C. taiwanensis* (LCA$^{Ct}$).

**Measures of similarity between genomes.** For each pair of genomes, we computed two measures of similarity, one based on gene repertoires and another based on the sequence similarity between two genomes. The gene repertoire relatedness was computed as the number of positional orthologs shared by two genomes divided by the number of genes in the smallest one[72]. Pairwise average nucleotide identities (ANIb) were calculated using the pyani Python3 module (https://github.com/widdowquinn/pyani), with default parameters[73]. We used single-linkage clustering to group strains likely to belong to the same species. This was done constructing a transitive closure of sequences with an ANIb higher than a particular threshold (i.e., >94, 95, or 96%). We used BioLayout Express$^{3D}$ to visualize the graphs representing the ANIb relationships and the resulting groups for each threshold (Supplementary Fig. 3).

**Inference of recombination.** We identified recombination events using three different approaches. We used the PHI test to look for incongruence within each core gene multiple alignment (Ct and C3 datasets). We made 10,000 permutations to assess the statistical significance of the results[74]. We used the SH-test, as implemented in IQ-TREE v1.3.8 [65] (GTR + I + G4 model, 1000 RELL replicates), to identify incongruence between the trees of each core gene and that of the concatenated multiple alignment of all core genes. We used ClonalFrameML v10.7.5 [75] to infer recombination and mutational events in the branch leading to the LCA$^{Ct}$ using the phylogenetic tree of C3 (Supplementary Data 2). The transition/transversion ratios given as a parameter to ClonalFrameML were estimated

with the R package PopGenome v2.1.6 [76]. Lastly, ClonalFrameML was also used to compare the relative frequency of recombination and mutation on the whole concatenated alignments of Ct and Rs.

**Molecular diversity and adaptation.** Positive selection was identified using likelihood ratio tests by comparing the M7 (beta)− M8 (beta&ω) models of codeml using PAML v4.8 [77]. We used the independent phylogenetic tree of each gene family to avoid problems associated with horizontal transfer (since many genes failed the SH-test for congruence with the core genome phylogenetic tree). We removed from the analysis gene families that had incongruent phylogenetic signals within the multiple alignment[78]. These correspond to the families for which PHI identified evidence of recombination ($P < 0.05$).

We inferred the mutations arising in the branch leading to LCA$^{Ct}$ using the phylogenetic tree build with the core genome of C3 (Ct and the five closest outgroups). First, we used ClonalFrameML to reconstruct the ancestral sequences of LCA$^{Ct}$ and LCA$^{C2}$ (accounting for recombination). Then, we estimated nucleotide diversity of each core gene for Ct, and between LCA$^{Ct}$ and LCA$^{C2}$ using the R package pegas. Finally, we used the branch-site model of codeml to identify positive selection on this branch for the core genes of C3 that lacked evidence of intragenic recombination (detected using PHI).

To infer the extent of purifying selection for Ct, we computed dN/dS values for each core genes between *C. taiwanensis* LMG19424 and the others strains of Ct using the yn00 model of PAML v4.8. We then plotted the average dN/dS of each strains with the patristic distances obtained from the tree of the concatenated multiple alignment of all core genes.

**Functional annotations.** We searched for the functions over-represented relative to a number of characteristics (recombination, nucleotide diversity, etc.). We analyzed COG categories, protein localizations, transporters, regulatory proteins, and several pre-defined lists of genes of interest in relation to rhizobial symbiosis and to virulence.

We used COGnitor[79] as available on the MicroScope Platform (https://www.genoscope.cns.fr/agc/microscope/home/) to class genes according to the COG categories (Supplementary Table 4 and Data 5). Protein subcellular localizations were predicted using PSORTb v3.0.2 (http://www.psort.org/psortb)[80]. Transporters and regulatory proteins were inferred using TransportDB (http://www.membranetransport.org/)[81] and P2RP (http://www.p2rp.org/)[82], respectively. Protein secretion systems were identified using TXSScan (http://mobyle.pasteur.fr/cgi-bin/portal.py#forms::txsscan)[83]. We manually checked and corrected the lists. Specific annotations were also defined for (i) *R. solanacearum* GMI1000: Type III effectors[84], PhcA-associated genes (i.e., genes involved in the upstream regulatory cascade controlling the expression of phcA, and genes directly controlled by PhcA)[38,85,86], virulence[87], extracellular polysaccharides (EPS)[86,88], chemotaxis[89], twin-arginine translocation pathway (Tat)[90], Tat-secreted protein[91], and (ii) the pRalta of *C. taiwanensis* LMG19424: symbiotic genes[19], genes pertaining to plasmid biology (conjugation, replication, partition, based on the annotations[92]), and operons using ProOpDB (http://operons.ibt.unam.mx/OperonPredictor/)[93]. Lastly, we also annotated positional orthologs between *R. solanacearum* GMI1000 and *C. taiwanensis* LMG19424 according to specific annotations used for both strains (Supplementary Table 4 and Data 5).

**Analysis of the mutations observed in the experimental evolution.** To estimate differences between mutation rates on the three replicons of the chimera, we compared the observed number of synonymous mutations in each replicon to those obtained from simulations of genome evolution. First, we analyzed the distribution of synonymous mutations of the 18 final evolved clones in regions of the genome that were covered by sequencing data (some regions with repeats cannot be analyzed without ambiguity in the assignment of mutations). We built the mutation spectrum of the genome using these synonymous mutations, since they are expected to be the least affected by selection. Second, we performed 999 random experiments of genome evolution using the mutation spectrum and the total number of synonymous mutations obtained for the 18 final clones. With the results, we draw the distributions of the expected number of synonymous mutations in each replicon (under the null hypothesis that they occurred randomly). These data were then used to define intervals of confidence around the average values observed in the simulations.

In addition, we used a similar approach to estimate the significance of parallelisms between evolved clones for SNP. We computed 99 random experiments of genome evolution using the mutation spectrum and the total number of SNP of each replicon for the 18 final clones and the three ancestors using a previously published method[17]. Then, we estimated how many nucleotide positions, genes, and COGs were affected in more than one clone for each random experiment. Lastly, we draw the distributions of the expected proportions of parallelisms to define intervals of confidence.

**Statistical analyses.** In order to identify genes that evolved faster in the branch leading to LCA$^{Ct}$, we compared the nucleotide diversity of sequences for LCA$^{Ct}$ and LCA$^{C2}$ with those of the extant 44 *C. taiwanensis* using a regression analysis.

Outliers above the regression line were identified using a one-sided prediction interval ($P < 0.001$) as implemented in JMP (JMP®, Version 10. SAS Institute Inc., Cary, NC, 1989–2007).

We computed functional enrichment analyses to identify categories over-represented in a focal set relative to a reference dataset. The categories that were used are listed above in the section "Functional annotations". To account for the association of certain genes to multiple functional categories, enrichments were assessed by resampling without replacement the appropriated reference dataset (see Supplementary Table 11) to draw out the expected null distribution for each category. More precisely, we made 999 random samples of the number of genes obtained for each analysis (positive selection, recombination, etc.) in the reference dataset. For each category, we then compared the observed value (in the focal set) to the expected distribution (in the reference dataset) to compute a $P$ value based on the number of random samples of the reference dataset that showed higher number of genes from the category.

We also compared the nucleotide diversity between sets of genes using the nonparametric Wilcoxon rank sum test ({stats}, wilcox.test).

Finally, we computed Fisher's exact tests (R package {stats}, fisher.test) to estimate the association between results of the natural and the experimental evolution, i.e., to test whether mutations found in the experimental evolution targeted genes that were found to be significantly more diverse in the natural process.

$P$ values were corrected for multiple comparisons using Benjamini and Hochberg's method[94] ({stats}, Pz.adjust).

Statistical analyses with R were done using version 3.1.3 (R: a language and environment for statistical computing, 2008; R Development Core Team, R Foundation for Statistical Computing, Vienna, Austria (http://www.R-project.org)).

**Data availability**. Genome sequence and annotation were made publicly available (GenBank BioProject: PRJEB23670, see accession numbers in Supplementary Data 3).

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

## Acknowledgements

This work was supported by funds from the French National Research Agency (ANR-12-ADAP-0014-01 and ANR-16-CE20-0011-01) the "Laboratoire d'Excellence (LABEX)" TULIP (ANR-10-LABX-41) and France Génomique National infrastructure, funded as part of "Investissement d'avenir" program managed by Agence Nationale pour la Recherche (contrat ANR-10-INBS-09). We thank Olaya Rendueles, Rémi Peyraud, Ludovic Cottret, Stéphane Genin, and Pedro Couto Oliveira for helpful comments and suggestions. We thank Eddy Ngonkeu and Moussa Diabate for help with the strain collection. We thank Céline Vandecasteele for the analysis of PacBio data. Sequencing was performed at the GeT core facility, Toulouse, France, supported in part by France Génomique National infrastructure (contract ANR-10-INBS-09).

## Author contributions

C.C., C.M.-B., and E.P.C.R. conceived the project, integrated the analyses, and wrote the draft of the manuscript. C.C., M.To., and E.P.C.R. made the computational analyses. D.C. and M.Ta. performed the experiments and analyzed the data. L.M. and M.A.P. provided

strains and data. C.G. and C.L.R. performed sequencing. S.C. assembled and annotated the genomes. All authors contributed to the final text.

## Additional information

**Competing interests:** The authors declare no competing interests.

