## [Peer Review File · Nature Communications]

Reviewers' comments:

Reviewer #1 (Remarks to the Author):

General comments

Authors identified adaptive changes during the integration of symbiosis functions in rhizobia associated with *Mimosa*. Experimental and natural evolution processes were compared despite great differences in the genome content of *Ralstonia solanacearum* and *Cupriavidus taiwanensis*. The chromosomal loci involved in the same quorum-sensing system other than those changes in loci on the symbiosis plasmid accounted for the parallel evolution processes. This work constitutes an important contribution to our understanding of the integration of symbiosis functions with genomic background, and more generally highlighted an effective way to study adaptive evolution underlying a complex trait by comparing experimental and natural evolution processes. The manuscript was well-written and this reviewer enjoyed reading it. However one major and several specific comments should be addressed.

Major comments

L181-183, and Figure 2, for those Ct strains without *imuABC* cassette, there are also less homologues to *pRalta-LMG19424* compared to other Ct strains harboring *imuABC*. Could this phenomenon result from transfer events independent of the acquisition of the plasmid harboring *imuABC*? Alternatively *imuABC* has been lost in these strains belonging to several sublineages of Ct. To clarify these possibilities, it would be good to know the similarity level between these symbiosis plasmids without *imuABC*.

L192-197, as to the effect of purifying selection, it would be more informative if the analysis was separately performed for Ct strains without *imuABC* and those with *imuABC*.

Likewise, in the proposed overall similarity between the experimental and natural evolutionary processes (Figure 5), the maintenance of *imuABC* might be just one of multiple options.

Specific comments

- (1) Figure 2, *sp.* should not be in italic.
- (2) L140, in the legend, Fig S8 should be reordered as Fig S2. Please also check the citation order of supplementary tables, such as Table S12 (L111)
- (3) L149, ", " should be deleted.
- (4) L164-168, the exact number of core genes (also for the frequency in the core genes of symbiosis plasmid) of incongruent phylogeny with the concatenated core genome tree should be shown. Similarly, more detailed information should be shown for the PHI test.
- (5) L171-172, and L522-526; 16S rDNA and 16 S should be replaced with 16S rRNA gene.
- (6) L211-212, to what extent recombination rates were lower for plasmid genes than the chromosomal ones.

Reviewer #2 (Remarks to the Author):

Clerissi et al, 2018 Nature Communications

Parallels between experimental and natural evolution of legume symbionts

General comments:

The manuscript by Clerissi and colleagues takes on the challenge of exploring parallels between the origins of symbiotic capacity in natural populations of rhizobia versus an in vitro evolution system in which symbiosis is initiated by the experimental horizontal transfer of a symbiosis plasmid and subsequent in planta selection.

This is an interesting approach and one that I have not seen before to compare genome changes in these two very different settings. The paper goes through a veritable mountain of data, with a variety of previously published and novel information. The bulk of the new information comes from genome sequencing (and or new genome analysis) of 60 strains of *Cupriavidus taiwanensis* as well as some fascinating experiments on different clones from the evolution experiment. The results largely fall into two parts: a) general genome patterns, where the authors compare the types and frequencies of mutations in the two different systems and b) test of specific loci and functions. For the latter, the most striking aspect of the paper are some competition experiments on mutants of Phc loci (that regulate cell density) showing that mutations that evolved in these loci improved infectivity when introduced into the ancestral strains.

While I find a lot of the data and discussion of this paper to be fascinating, I find the organization and approach to be frustrating in key ways. First and foremost, much of the manuscript has more of a 'book chapter' feel than that of an empirical paper. Rather than clearly laying out specific hypotheses, predictions, and tests, the paper synthesizes different sets of disparate data and reviews data that are consistent with the overarching hypothesis that the natural and experimental systems are 'parallel'. However, this overarching hypothesis is not really treated in a scientific or rigorous way, since the authors only review patterns that they see as similar between the natural and experimentally evolved rhizobia. There are few explicit tests of this hypothesis and rarely is there any clear null hypothesis. In other words, there are both genomic similarities and differences that occurred between the natural and experimental evolution of rhizobia, but since the paper only focused on similarities it ends up providing a rather one sided view of the topic.

Specific comments:

1. Lines 40-42. The introductory sentence might seem naïve to many readers, since adaptation is not only studied using a comparative (evolutionary) approach. This is only one method, and this ignores the rich history of experimental and population analyses of fitness variation that is used to study adaptation.
2. Line 47 (and elsewhere) The word data is a plural.
3. Lines 51-57. This section worries me as it presents a rather biased view that horizontal transfer of novel loci is the key step in adaptation to novel environments. Clearly there is

some importance to this step, but data herein and elsewhere shows that the rest of the genome might be even more important. Also, this portion assumes a temporal ordering that might not be correct.

4. The introduction is wholly lacking in hypotheses and I think that this is where the manuscript could be greatly strengthened in terms of providing an organized structure and balanced view of the data.

5. Lines 96-101 present data on SNP and deletions are presented without any context. What would be the expectation for SNPs and or deletions? The data that are described in both of these contexts are fully consistent with drift (random association of SNPs and association of deletions with TEs). The reader could easily misconstrue this section.

6. Lines 111-113. These data need to be better explained. What precisely is meant by shared events. Does this mean SNPs in the same base? The same base changes? Some of these data are much more important than others. The fact that the same COG category has a change might not mean anything at all.

7. Lines 169-173. The data here, that the three rhizobial clades evolved independently, is anecdotal. Instead the authors should test this hypothesis by doing a likelihood ratio test in which they compare a constrained tree to an unconstrained one (or another similar test).

8. Lines 178-180. There is not sufficient explanation of the imu plasmid cassette. The authors cannot assume that the reader understands this context.

9. Lines 181-184. This argument is not supported by any data and should be dropped. Just because the imu cassette is in most of the strains does not mean that it played an important role.

10. Lines 189-192. These results need to be put into context. A similar set of genes evolved more in both systems, but isn't that the null expectation? Why would we have any other expectation?

11. Lines 192-197 As above, there is evidence of purifying selection on a set of core loci in both systems. As above, this must be the null model. Of course there is going to be a set of key genes for organismal function that will be under purifying selection in many/most environments.

12. Lines 231-233. I found the use of the term 'diversification' to be somewhat awkward in this manuscript, and especially here where diversification and positive selection are equated. Perhaps the authors would be better to use 'adaptation' instead of diversification?

13. Line 339. Why is gene expression discussed here. I could not find any analysis that focused on expression data.

14. Line 341. What do the authors mean by the 'cost' of the plasmid. This does not seem to match any data that they have tested. Please explain.

Reviewer #3 (Remarks to the Author):

Review of the manuscript by Clerissi et al. titled "Parallels between experimental and natural evolution

In this work the authors have carried out a double experimental approach to characterize the nature of an important symbiogenetic phenomenon: the evolution of legume

symbionts. One first study, that can be considered as of classical nature, the authors characterize by comparative genomics the evolution of symbiosis in the genus *Cupriavidus*. In the second experiment, the authors have carried out an experimental evolution by introducing a symbiotic plasmid of *Cupriavidus* in *Ralstonia* and tested the capability of this bacteria to nodulate and to fix nitrogen in the *Mimosa* host, it being the same that in *Cupriavidus*. The authors have also carried out many comparative genomic analyses and also additional experiments in *Ralstonia* to test the selective role played by different genes, either from the plasmid or from the main chromosome of *Ralstonia*.

Main point

The authors are aware that time and environmental conditions are two critical factors that make problematic the comparison between both systems. However, they have been able to provide a number of commonalities between natural and experimental evolution of symbiosis. Maybe the authors were expecting to find a case in *Ralstonia* of either nodulation and nitrogen fixation for later to carry out a detailed analysis of how the symbiosis was finally established from an initial pathogen. But this has not been the case; the authors have found three chimeric clones out of 16 able to nodulate, but not a single case of nitrogen fixation in the experimental evolution. Is it a matter of time or a question regarding the complex nature of the environment that is not really mimicked in the experimental evolution? As it happens many times in evolution, evidence of processes derived from comparative phylogenetics of evolving traits are not reproducible at the level of real time experimental evolution, something that is clearly stated and cited by the authors.

The work carried out, however, is impressive, particularly the detailed analyses performed at comparative genomic level. As the authors detail there are some lessons that can be derived after a "proper" comparison between real time experimental evolution and natural evolution.

Comments

1) A missing point in this study is to bring some light on an eternal debate on the relative role played by positive (natural) selection versus other forces in the evolution of symbiosis. The authors state that plasmid evolution in both systems are under purifying selection. However, when performing analysis on different nuclear genes in both systems (lines 236-238 and 247-252), the authors apparently conclude that adaptation is taking place when the symbionts are evolving in the *Mimosa* host. This question should be discussed in order to locate the work in a more general register. In other words: is natural selection a driving force in the symbiotic process? The authors consider that their work is not conclusive (lines 369-374), when probably they have more conclusive results that they do not dare to conclude.

2) Something should be said or analyses to be done on the role played by mobile genetic elements. No evidence of MGE and gene inactivations, particularly in *Cupriavidus*? The authors recognize its importance (lines 346-349), but such analyses are lacking.

3) Statement on lines 375-377 is very vague. What is the meaning of: "high-order analyses"?

Answers to the Reviewers

Reviewer 1

General comments

*Authors identified adaptive changes during the integration of symbiosis functions in rhizobia associated with Mimosa. Experimental and natural evolution processes were compared despite great differences in the genome content of *Ralstonia solanacearum* and *Cupriavidus taiwanensis*. The chromosomal loci involved in the same quorum-sensing system other than those changes in loci on the symbiosis plasmid accounted for the parallel evolution processes. This work constitutes an important contribution to our understanding of the integration of symbiosis functions with genomic background, and more generally highlighted an effective way to study adaptive evolution underlying a complex trait by comparing experimental and natural evolution processes. The manuscript was well-written and this reviewer enjoyed reading it.*

We thank the Reviewer for the encouragement to re-submit our work and pointing out the relevance of the topic.

However one major and several specific comments should be addressed.

Major comments

Comment n° 1.1:

L181-183, and Figure 2, for those Ct strains without imuABC cassette, there are also less homologues to pRalta-LMG19424 compared to other Ct strains harboring imuABC. Could this phenomenon result from transfer events independent of the acquisition of the plasmid harboring imuABC? Alternatively imuABC has been lost in these strains belonging to several sublineages of Ct. To clarify these possibilities, it would be good to know the similarity level between these symbiosis plasmids without imuABC.

Answer to comment n° 1.1:

There are three hypotheses.

1) Different symbiotic plasmids were acquired in different strains. This does not fit the data.

- The genes in the nod-nif locus are highly similar between all plasmids. The similarity between these genes is as high as that of the chromosomal core genes in agreement with the hypothesis of vertical descent.

- The analysis using birth-death models suggests one single integration of these genes in the lineage.

- Overall, the core genes in pRALTA show fewer, not more, phylogenetic incongruence with the core genome phylogenetic tree than the chromosomal core genes, suggesting that they were vertically inherited.

2) ImuA2B2C2 was acquired after pRALTA and only in some strains. But:

- The level of similarity between ImuA2B2C2 genes matches the distribution of similarity among core genes, suggestive of vertical inheritance.

- The only exception to the previous point concerns the genes from the clade containing STM6041, where the PacBio sequence shows an additional plasmid with an homologous imuABC that has lower sequence similarity to the ImuA2B2C2 of pRALTA.

- In spite of this exception, the analysis using birth-death models suggests one single integration of the pRALTA ImuA2B2C2 genes in the lineage (Table S3).

3) Multiple deletions of ImuA2B2C2.

- This scenario fits the above data. Given the large number of transposable elements in the plasmid, such deletions are not unlikely.

Action. We now show the data on birth-death models for the acquisition of the rhizobial trait (Fig. S4). We also show the percent similarity of imuA2B2C2 proteins and those of the nod-nif locus while showing the same analysis for all core proteins for comparison (Figures S5 and S6). We changed the text to emphasize the above arguments (see also comment 2.1).

Figure S5: Percent similarity between ImuA2B2C2 proteins in *C. taiwanensis* LMG19424 and the remaining *C. taiwanensis* genomes. On the right, we present the same analysis for all core proteins. The points below 90% similarity in the first three graphs correspond to the clade STM6041 where (at least in this genome) the locus is carried by another plasmid.

Figure S6: Percent similarity between the genes in pRALTA nod-nif locus and similar representation for all core proteins. With one single exception (pRALTA_0388), all proteins of the locus had more than 94% similarity with their orthologs in all other genomes of the species.

Comment n° 1.2:

L192-197, as to the effect of purifying selection, it would be more informative if the analysis was separately performed for Ct strains without imuABC and those with imuABC.

Answer to comment n° 1.2:

We have colored the points in the supplementary figure S8 in function of presence of the cassette in the comparison with the reference strain. This shows similar patterns of purifying selection and few differences between the strains.

Comment n° 1.3:

Likewise, in the proposed overall similarity between the experimental and natural evolutionary processes (Figure 5), the maintenance of imuABC might be just one of multiple options.

Answer to comment n° 1.3:

We have modified the text to "Continued and frequent presence of *imuA2B2C2* cassette. This makes the text less assertive and more open to other possible mechanisms (and more factual). This also answers to comment 2.9.

Specific comments

Comment n° 1.4:

(1) Figure 2, sp. should not be in italic.

Answer to comment n° 1.4:

Thank you. We have made the modification.

Comment n° 1.5:

(2) L140, in the legend, Fig S8 should be reordered as Fig S2. Please also check the citation order of supplementary tables, such as Table S12 (L111)

Answer to comment n° 1.5:

We checked and corrected all figures and tables numbers.

Comment n° 1.6:

(3) L149, "," should be deleted.

Answer to comment n° 1.6:

This was corrected.

Comment n° 1.7:

(4) L164-168, the exact number of core genes (also for the frequency in the core genes of symbiosis plasmid) of incongruent phylogeny with the concatenated core genome tree should be shown. Similarly, more detailed information should be shown for the PHI test.

Answer to comment n° 1.7:

This information was added to the text (l. 187 and 190) (it was previously available in Table S3).

Comment n° 1.8:

(5) L171-172, and L522-526; 16S rDNA and 16 S should be replaced with 16S rRNA gene.

Answer to comment n° 1.8:

This was corrected.

Comment n° 1.9:

(6) L211-212, to what extent recombination rates were lower for plasmid genes than the chromosomal ones.

Answer to comment n° 1.9:

We have added the values of $(\text{Observed}-\text{Expected})/(\text{Observed}+\text{Expected})$ associated with the analysis of PHI and SH.

Reviewer 2

General comments:

The manuscript by Clerissi and colleagues takes on the challenge of exploring parallels between the origins of symbiotic capacity in natural populations of rhizobia versus an in vitro evolution system in which symbiosis is initiated by the experimental horizontal transfer of a symbiosis plasmid and subsequent in planta selection.

This is an interesting approach and one that I have not seen before to compare genome changes in these two very different settings.

We thank the Reviewer for the appreciation of our work.

*The paper goes through a veritable mountain of data, with a variety of previously published and novel information. The bulk of the new information comes from genome sequencing (and or new genome analysis) of 60 strains of *Cupriavidus taiwanensis* as well as some fascinating experiments on different clones from the evolution experiment. The results largely fall into two parts: a) general genome patterns, where the authors compare the types and frequencies of mutations in the two different systems and b) test of specific loci and functions. For the latter, the most striking aspect of the paper are some competition experiments on mutants of *Phc* loci (that regulate cell density) showing that mutations that evolved in these loci improved infectivity when introduced into the ancestral strains.*

Comment n° 2.1:

While I find a lot of the data and discussion of this paper to be fascinating, I find the organization and approach to be frustrating in key ways. First and foremost, much of the manuscript has more of a 'book chapter' feel than that of an empirical paper. Rather than clearly laying out specific hypotheses, predictions, and tests, the paper synthesizes different sets of disparate data and reviews data that are consistent with the overarching hypothesis that the natural and experimental systems are 'parallel'. However, this overarching hypothesis is not really treated in a scientific or rigorous way, since the authors only review patterns that they see as similar between the natural and experimentally evolved rhizobia. There are few explicit tests of this hypothesis and rarely is there any clear null hypothesis. In other words, there are both genomic similarities and differences that occurred between the natural and experimental evolution of rhizobia, but since the paper only focused on similarities it ends up providing a rather one sided view of the topic.

Answer to comment n° 2.1:

- Lack of explicit hypothesis and tests

Answer. The manuscript summarizes a very large set of data and included many thousands tests of hypothesis treated in a scientific and rigorous way using state-of-the-art methods. We agree that the writing tended to obscure this fact because the sheer number of tests had led us to include this information in supplementary material.

Action. We have changed the text in the results section to make hypotheses more explicit. We brought to the main text more statistical tests.

- One-sided view of the process.

Answer. We acknowledge the way we presented the paper may appear as a one sided view of the process. However:

We did compare the two processes and showed differences and similarities. We mentioned differences between the two processes (eg the analysis of the T3SS), and cases where parallels could not be analyzed (e.g., hyper-mutagenesis, specific *R. solanacearum* or *C. taiwanensis* genes).

We specifically focused the discussion on the parallels because, in our opinion, this was the key novel contribution of our study.

Indeed, there were major differences in terms of time-span, environment, genetic background and phenotypic achievement between the two processes, (cf abstract and introduction). We thus expected many differences and few - if any - similarities. In addition, many previous works have shown that there are differences between apparently similar adaptive processes (and we confirm that).

To compare the two processes, we focused on some specific aspects. This was not to give a biased view of the similarities. Instead, the previous experimental work showed specific evolutionary changes and patterns (selection type, adaptive loss of T3SS, adaptive regulatory rewiring, role of a mutagenesis cassette) and raised specific questions (plasmid versus chromosomal adaptation). This guided our analysis throughout the paper.

Action. We have modified the introduction (final paragraph) to better explain our approach and present the aspects we focused on. We have better explained our hypotheses in the Result section. We have also emphasized that many changes are not parallel (not even between runs of the experiment).

Specific comments:

Comment n° 2.2:

Lines 40-42. The introductory sentence might seem naïve to many readers, since adaptation is not only studied using a comparative (evolutionary) approach. This is only one method, and this ignores the rich history of experimental and population analyses of fitness variation that is used to study adaptation.

Answer to comment n° 2.2:

We did not intend to exclude other aspects of evolutionary biology, just focus on these two. We have modified the introductory sentence accordingly.

Comment n° 2.3:

Line 47 (and elsewhere) The word data is a plural.

Answer to comment n° 2.3:

This was corrected.

Comment n° 2.4:

Lines 51-57. This section worries me as it presents a rather biased view that horizontal transfer of novel loci is the key step in adaptation to novel environments. Clearly there is some importance to this step, but data herein and elsewhere shows that the rest of the genome might be even more important. Also, this portion assumes a temporal ordering that might not be correct.

Answer to comment n° 2.4:

The sentence did not exclude other factors, it just emphasized the frequent role of HGT in adaptation to new environments. We have changed the sentence to focus on the importance of horizontal transfer in transitions towards symbiosis (not adaptation in general) and to highlight that there are many examples of this (but we do not exclude that adaptation can occur by other means). We also added that successful adaptation upon transfer requires specific genetic backgrounds (hence, some changes in the genome may be required prior to transfer).

Comment n° 2.5:

The introduction is wholly lacking in hypotheses and I think that this is where the manuscript could be greatly strengthened in terms of providing an organized structure and balanced view of the data.

Answer to comment n° 2.5:

We have changed the last paragraph to better explain our goal and present the specific aspects on which we focused our analysis. See also answer to comment 2.1.

Comment n° 2.6:

Lines 96-101 present data on SNP and deletions are presented without any context. What would be the expectation for SNPs and or deletions? The data that are described in both of these contexts are fully consistent with drift (random association of SNPs and association of deletions with TEs). The reader could easily misconstrue this section.

Answer to comment n° 2.6:

This comment is linked with comment 2.7. We now provide statistical tests taking into account the expected distributions of mutations (SNPs) if they occurred at random. This shows that the frequency with which mutations appear in the same positions is higher than expected by a model of random assortment of mutations. As for the deletions, they cannot have occurred at random since most of them are in the same locations of the plasmid (note that since we show that plasmid mutations are not adaptive we do not find pertinent to detail the results concerning these large deletions).

Comment n° 2.7:

Lines 111-113. These data need to be better explained. What precisely is meant by shared events. Does this mean SNPs in the same base? The same base changes? Some of these data are much more important than others. The fact that the same COG category has a change might not mean anything at all.

Answer to comment n° 2.7:

-What is meant by shared events?

Answer. The bar indicates the number of parallel experiments where the exact same position has been mutated several times. The bar on genes indicates that the same for genes, etc.

Action. “Shared events” has now been clarified in the text and in the legend of figure 1.

-COG analysis is meaningless.

Answer. Analysis using COG categories is standard in the genomics literature where it provides coarse-grained information on functional differences. Note that very similar types of analyses at broader functional units have been published before also in the experimental evolution literature (Tenaillon, Science, 12).

Action. We have now indicated the p-value of the test concerning the COG classes.

-Different levels of information.

Answer. The analysis is done on purpose to analyze parallelisms at different levels of granularity. This was unclear (see answer to reviewer #3.5).

Action. We now show that parallelisms at nucleotide, gene and COG levels are statistically significant. We changed the text to make clear that this provides different levels of information.

Comment n° 2.8:

Lines 169-173. The data here, that the three rhizobial clades evolved independently, is anecdotal. Instead the authors should test this hypothesis by doing a likelihood ratio test in which they compare a constrained tree to an unconstrained one (or another similar test).

Answer to comment n° 2.8:

We aimed at testing if the presence of the rhizobial genes in the three clades can be explained by one single integration of these genes or requires several independent integrations. For this, one has to assess two levels of uncertainty. First, uncertainty in the phylogenetic reconstruction. As indicated in Fig. S2, most nodes in the *Cupriavidus* tree have 100% bootstrap support (all bootstrap values are given in Table S18). The three clades are clearly separated in the tree by several nodes with 100% bootstrap and the comparison between a constrained and an unconstrained tree is not necessary. The second source of uncertainty concerns the inference of the ancestral values in terms of the presence of the rhizobial-associated genes (the core genes in the *nod-nif* locus). We insist that this is not anecdotal, it is essential to make the claim. The analysis was done using rigorous birth-death processes that are the current state-of-the-art for this type of analysis. All results are consistent with a single initial acquisition of the plasmid in this clade”)

Action. We have now indicated in the text the information on the bootstrap values of the tree. We have also added a novel supplementary figure showing for *nif-nod* core genes, the probability that they were horizontally acquired in the three branches (see also comment #1.1). As can be seen from this distribution, the values are very high for nearly all genes (all but one are above the standard cut-off of 50% for this type of analysis).

Figure S4. The distribution of the probability of horizontal acquisition of *nif-nod* genes in the three branches indicated in the phylogenetic tree. With exception of one gene in two cases, all genes have a probability higher than 50% of being acquired in the branches by HGT.

Comment n° 2.9:

8. Lines 178-180. There is not sufficient explanation of the *imu* plasmid cassette. The authors cannot assume that the reader understands this context.

Answer to comment n° 2.9:

We have now provided additional information on the *imuA2B2C2* cassette and its role in the experimental symbiotic evolution of *R. solanacearum*.

Comment n° 2.10:

9. Lines 181-184. This argument is not supported by any data and should be dropped. Just because the *imu* cassette is in most of the strains does not mean that it played an important role.

Answer to comment n° 2.10:

We were very cautious as we wrote “Yet, we were able to identify the *imuA2B2C2* cassette in most extant strains, suggesting that they could have played a role in the symbiotic evolution of *Cupriavidus*”.

In response to Reviewer 1 (comment #1.3), we evaluated the possibility that symbiotic plasmids with the cassette were independently acquired from those without the cassette. Data argued in favor of a unique acquisition of an ancestral pRalta symbiotic plasmid in the *Cupriavidus taiwanensis* branch, followed by several losses of the cassette.

We have now added this information and modified the text.

Comment n° 2.11:

10. Lines 189-192. These results need to be put into context. A similar set of genes evolved more in both systems, but isn't that the null expectation? Why would we have any other expectation?

Answer to comment n° 2.11:

Answer. We may have explained this poorly (or too succinctly). We don't compare the genetic diversity in the branch to the number of mutations in the experiment. This could reveal a parallelism but could also result from general trends in terms of the rate of evolution of genes (many genes evolve rapidly in any situation and many genes are always highly conserved). What we do is to compare the *excess* (in the text: "Clones of the evolution experiment accumulated significantly more mutations in genes whose orthologs had an excess of polymorphism at the onset of symbiosis in natural populations") of genetic diversity in the branch bLCA^{Ct} *relative* to the genetic diversity of the gene in the species. The genes in this analysis that reject the null model (number of changes in the branch proportional to the gene diversity in the species) have endured an accelerated evolution in the branch. Then we took these genes and find that they over-represent genes with mutations in the experiment. Here, the null hypothesis is that the two sets (genes with excess of diversity in the branch bLCA^{Ct} and genes with mutations in the experiment) are independent.

Action. We have changed the text to clarify the null hypothesis and the method.

Comment n° 2.12:

Lines 192-197 As above, there is evidence of purifying selection on a set of core loci in both systems. As above, this must be the null model. Of course there is going to be a set of key genes for organismal function that will be under purifying selection in many/most environments.

Answer to comment n° 2.12:

Answer. Purifying selection is indeed expected in natural populations but not in experimental evolution (when there is adaptation). Following are a few examples: 1) In the first 20,000 generations of *E. coli* long-term evolution (LEE), all substitutions in genes are non-synonymous (Barrick, Nature, 09); 2) Paul Rainey's work on bet-hedging in *Pseudomonas fluorescens* (Beaumont, Nature, 09) found 9 mutations in genes, all non-synonymous. 3) Michael Brockhurst paper on antagonistic co-evolution in *Pseudomonas fluorescens* does not give exact numbers, but states that very few non-synonymous substitutions were identified (Paterson, Nature, 10). 4) Olivier Tenaillon and Brandon Gaut paper on parallels in experimental evolution of *E. coli* (Tenaillon, Science, 12) find six times more non-synonymous than synonymous changes (twice the expected number). 5) The team of Genin (MBE, 14) showed adaptive experimental evolution of *Ralstonia solanacearum* and 8 out 9 mutations were non-synonymous. 6) Finally, Isabel Gordo and colleagues (Perfeito, Science, 07) show that on a period of adaptation of *E. coli* there is a large supply of adaptive mutations that interfere to reach fixation and this drives the fixation of mutations, with the consequence that most mutations fixating under these conditions are adaptive (thus most often non-synonymous).

Action. We have now clarified these points in the text (and added references).

Comment n° 2.13:

Lines 231-233. I found the use of the term 'diversification' to be somewhat awkward in this manuscript, and especially here where diversification and positive selection are equated. Perhaps the authors would be better to use 'adaptation' instead of diversification?

Answer to comment n° 2.13:

The sentence reads " To evaluate whether the observed rapid plasmid diversification was driving the adaptation to symbiosis *in natura* , ...". If we replace diversification by adaptation the sentence becomes false: the plasmid does not adapt. It diversifies in a non-adaptive way. We use the word "diversification" for genetic diversification, i.e., a process leading to the increase in genetic diversity. We have systematically added "genetic" to diversification to make its meaning clear.

Comment n° 2.14:

Line 339. Why is gene expression discussed here. I could not find any analysis that focused on expression data.

Answer to comment n° 2.14:

We have removed "gene expression or biochemical fine-tuning".

Comment n° 2.15:

Line 341. What do the authors mean by the 'cost' of the plasmid. This does not seem to match any data that they have tested. Please explain.

Answer to comment n° 2.15:

Answer. The pRalta plasmid does not have a fitness cost *ex planta* in the original chimera. This was indicated in Table S11 (previously in lines 245-247). We now add a supplementary figure (Fig. S9) with the data and the appropriate statistical tests.

Figure S9. Analysis of the cost of the pRalta plasmid in *R. solanacearum* GMI1000. Percentage of alive bacteria *ex planta* in Jensen medium (top) or in Jensen-Mimosa medium (bottom). The values correspond to the last day of the experiment for which all intermediate data is available in Table S11 (point at day 22). Each experiment was replicated 12 times. The clone with the plasmid has slightly higher survival rates (indicating a benefit instead of a cost), but the differences are not statistically significant. Means diamonds indicate at the center the mean, and at the edges the 95% confidence interval.

Reviewer 3

*In this work the authors have carried out a double experimental approach to characterize the nature of an important symbiogenetic phenomenon: the evolution of legume symbionts. One first study, that can be considered as of classical nature, the authors characterize by comparative genomics the evolution of symbiosis in the genus *Cupriavidus*. In the second experiment, the authors have carried out an experimental evolution by introducing a symbiotic plasmid of *Cupriavidus* in *Ralstonia* and tested the capability of this bacteria to nodulate and to fix nitrogen in the *Mimosa* host, it being the same that in *Cupriavidus*. The authors have also carried out many comparative genomic analyses and also additional experiments in *Ralstonia* to test the selective role played by different genes, either from the plasmid or from the main chromosome of *Ralstonia*.*

Comment n° 3.1:

*The authors are aware that time and environmental conditions are two critical factors that make problematic the comparison between both systems. However, they have been able to provide a number of commonalities between natural and experimental evolution of symbiosis. May be the authors were expecting to find a case in *Ralstonia* of either nodulation and nitrogen fixation for later to carry out a detailed analyses of how the symbiosis was finally established from an initial pathogen. But this has not been the case; the authors have found three chimeric clones out of 16 able to nodulate, but not a single case of nitrogen fixation in the experimental evolution.*

Answer to comment n° 3.1:

Answer. In a first “selection step”, where the chimeric ancestor GMI1000pRalta (unable to nodulate) was inoculated to hundreds of *Mimosa pudica* plantlets, we isolated 3 mutants able to nodulate *M. pudica*. These three clones were further evolved by serial cycles of inoculation to *M. pudica* and re-isolation of bacteria from nodules. 18 lines were derived in parallel and after 16 cycles a clone isolated from each population was analyzed. All clones isolated in cycle 16 nodulate *M. pudica* (not just three) but none fixes nitrogen in symbiosis with *M. pudica*.

Action. The experimental setting and main phenotypes are given in figure 1. The first section of the results explicitly states that mutualistic nitrogen fixation was not observed after 16 cycles.

Comment n° 3.2:

Is it a matter of time or a question regarding the complex nature of the environment that is not really mimicked in the experimental evolution? As it happens many times in evolution, evidence of processes derived from comparative phylogenetics of evolving traits are not reproducible at the level of real time experimental evolution, something that is clearly stated and cited by the authors.

Answer to comment n° 3.2:

We agree that at this stage it is not possible to decide between the two.
We did not modify the text.

The work carried out, however, is impressive, particularly the detailed analyses performed at comparative genomic level. As the authors detail there are some lessons that can be derived after a “proper” comparison between real time experimental evolution and natural evolution.

We thank the Reviewer for his/her appreciation of our work.

Comment n° 3.3:

A missing point in this study is to bring some light on an eternal debate on the relative role played by positive (natural) selection versus other forces in the evolution of symbiosis. The authors states that plasmid evolution in both systems are under purifying selection. However, when performing analysis on different nuclear genes in both systems (lines 236-238 and 247-252), the authors apparently conclude that adaptation is taking place when the symbionts are evolving in the Mimosa host. This question should be discussed in order to locate the work in a more general register. In other words: is natural selection a driving force in the symbiotic process? The authors consider that they work is not conclusive (lines 369-374), when probably they have more conclusive results that they do not dare to conclude.

Answer to comment n° 3.3:

Answer. We believe there are two different concepts of selection here, and our text may have been ambiguous.

Plants select their rhizobial bacteria. This is determined in part by the dialogue between the plant and the bacterium. It is possible that the regulatory re-wiring we observe (associated with the evolution of the *phc* genes) is related with this dialogue. At this stage, conclusions must wait for further experimental work. Natural selection in the experiment. We believe this is the sense given by the reviewer to selection. We agree with the reviewer that we can make a better case for the action of natural selection in this system. We do identify cases of positive selection and we do show them to be adaptive in the lab. Hence, positive selection seems to play an important role in the symbiosis.

Action. We have added a sentence in the introduction to mention the problem of selection by the plant. We have changed the text in the discussion to tackle this point. We also added a mention to positive selection in the abstract.

Comment n° 3.4:

*Something should be said or analyses to be done on the role played by mobile genetic elements. No evidence of MGE and gene inactivations, particularly in *Cupriavidus*? The authors recognize its importance (lines 346-349), but such analyses are lacking.*

Answer to comment n° 3.4:

Answer. We are not certain of understanding this point. A significant part of the study is focused on the evolution of the plasmid (an MGE). Also, many of the results concern recombination and horizontal transfer in *Ct*, which are usually mediated by MGE.

Maybe the reviewer refers specifically to transposable elements (IS)? We mention in the first section of the results that most deletions take place in the plasmid and are flanked by transposable elements.

To analyze transpositions precisely, we would need completely assembled genomes both for the clones resulting from the experiments and for the natural isolates (more than 50 genomes should be sequenced). This is not available, and sequencing them by PacBio would be very costly.

To assess if such investment was worthwhile, we took the short-read data of half of the clones in the experiment and searched for IS transposition using a recently published tool (ISmapper). We could only observe six confident events of transposition (low "gap values" given by ISmapper), none of which parallel. Since this analysis does not seem very promising, and would be much more difficult in the more divergent natural isolates (for which the reference fully assembled genomes are much more divergent than in the experience), we have not pursued our efforts any further.

Action. We show the analysis below. We don't think this analysis is sufficiently interesting to be included in the manuscript.

Methods. To identify IS on the *Ralstonia* chimera, we searched the replicons for homologs in the database ISFinder (<http://www-is.biotoul.fr/>) (Siguier et al. 2006. Nucleic Acids Research) using blastp (evalue<10⁻⁵). In order to identify IS transpositions, we searched for the IS sequences of the genomes on the chimera replicons using blastp (evalue<10⁻⁵). We downloaded sequences of best blastp hits, and we searched for similar sequences in the nucleotide sequences of the replicons using blastn. Putative IS were kept only if the identity was high (>80%), the alignment length was longer than 100 nucleotides and the evalue<10⁻⁵. We computed a blastn of the chimera IS database against itself to kept only one representative IS for each group of very similar elements (same parameters). Then 27 representative ISs were analyzed with ISMapper (Hawkey et al. 2015. BMC Genomics) on the collections of paired-end sequencing dataset of 9 final evolved clones. The ISMapper typing option was used to find IS query locations in short read data, and to compare these locations to the reference genome of the chimera.

Results. The following table shows the putative novel insertions in the genome.

Table: Results of the analysis of ISMapper on the sequences of nine final clones of the experiment. The positions *From* and *To* indicate the positions of the insertion in the reference genome of *R. solanacearum* GMI100 (chromosomes 1 and 2) and pRALTA of *C. taiwanensis* LMG19424.

Clones	IS	Replicon	Novel insertion	
			From	To
J16	ISRta1	Chromosome 1	20848	20859
J16	ISRso9	Chromosome 1	2716086	2716096
E16	ISRso10	Chromosome 2	708487	708499
F16	ISRso5	Chromosome 2	1011023	1011028
F16	ISRta1	Chromosome 2	1564552	1564563
T16	ISBph3	pRalta	223502	223496

Comment n° 3.5:

Statement on lines 375-377 is very vague. What is the meaning of: “high-order analyses”?

Answer to comment n° 3.5:

We meant that mutations are seldom the same between replicates. But more often they target the same gene or group of genes in different replicates. We have revised the sentence for clarity (see also comment 2.7).

REVIEWERS' COMMENTS:

Reviewer #1 (Remarks to the Author):

This reviewer had a major concern about those strains lacking imuABC, which were not very effectively considered in the proposed general model in the previous version of the manuscript. In this revision, additional analyses have been added and multiple deletions of imuABC is considered in the proposed general model. As described below, there are still some minor concerns, which could be easily addressed by authors.

(1) In this revision, birth-death models for the acquisition of symbiosis genes (Fig S4) together with other evidences suggest a single integration of these genes in the lineage. Percent similarity between imuABC genes matches the distribution of similarity among core genes suggesting a vertical evolution (Fig S5). Birth-death models suggest one single integration of the pRALTA imuABC genes in the lineage (Table S3 Core genomes and corresponding phylogenetic trees for the different clades.). But this information is not immediately available to readers in the cited Table S3.

(2) Percent similarity between the genes in pRALTA nod-nif locus is presented (Fig S6). The results of 22 genes are shown in Fig S6. Are they exactly all homologues shared between Ct genomes? If this reviewer got it right (Table S6), there are 51 core genes of pRALTA shared by test Ct genomes. The information for all 51 genes/proteins should be presented or summarized. This would make the proposed model of multiple deletions of imuABC more convincing.

Specific comments:

1) In the legend of Fig S4, "With exception of one gene in two cases, ..." It would be easier for readers to follow to show the gene name herein. "sp." in the phylogenetic tree of Fig S4 should not be in italic.

2) Different fonts are present in the legend of Fig S5.

3) The figure legend of Fig S6 (95% similarity) is not consistent with that shown in "Answers to Reviewers" (94% similarity). It is supposed to be 94% if NifX-like, NodS and NifQ were included. Fig S6, It should be clearly indicated that this analysis was performed for Ct genomes.

Reviewer #2 (Remarks to the Author):

The authors have very carefully dealt with the critical comments of my review and the other reviews as well. I am happy with the current version of the manuscript as it stands.

Reviewer #3 (Remarks to the Author):

This new version takes into account all my comments as well as those made by the other reviewers.

Answer to Reviewer #1:

(1) In this revision, birth-death models for the acquisition of symbiosis genes (Fig S4) together with other evidences suggest a single integration of these genes in the lineage. Percent similarity between imuABC genes matches the distribution of similarity among core genes suggesting a vertical evolution (Fig S5). Birth-death models suggest one single integration of the pRALTA imuABC genes in the lineage (Table S3 Core genomes and corresponding phylogenetic trees for the different clades.). But this information is not immediately available to readers in the cited Table S3.

The reviewer is correct, but the data is already plotted in Sup. Figs 1 and 5. No changes made to the text.

(2) Percent similarity between the genes in pRALTA nod-nif locus is presented (Fig S6). The results of 22 genes are shown in Fig S6. Are they exactly all homologues shared between Ct genomes? If this reviewer got it right (Table S6), there are 51 core genes of pRALTA shared by test Ct genomes. The information for all 51 genes/proteins should be presented or summarized. This would make the proposed model of multiple deletions of imuABC more convincing.

Our text stated precisely that this analysis concerned the *imuABC* and the core *nod-nif* genes not all core genes in the plasmid. We focused on these genes because the reviewer questioned our statement that the traits had been integrated only once. This analysis answers precisely to this point (along with the other cited analyses). We could put the other 30 histograms in supplementary material if strictly required by Nature Communications but they are not relevant to answer to this question and their interest is very limited (they look alike). Please note that the high similarity among these families of 51 core genes is already shown in Figure 3B (it's indicated as "diversity" and is always lower than 10%).

Specific comments:

1) In the legend of Fig S4, "With exception of one gene in two cases, ..." It would be easier for readers to follow to show the gene name herein. "sp." in the phylogenetic tree of Fig S4 should not be in italic.

The gene name is now indicated: *noIG*.

The figure was corrected.

2) Different fonts are present in the legend of Fig S5.

This is now correct.

3) The figure legend of Fig S6 (95% similarity) is not consistent with that shown in "Answers to Reviewers" (94% similarity). It is supposed to be 94% if NifX-like, NodS and NifQ were included. Fig S6, It should be clearly indicated that this analysis was performed for Ct genomes.

This was corrected (it was indeed 94%). The information on Ct was added.

Reviewer #2 (Remarks to the Author):

The authors have very carefully dealt with the critical comments of my review and the other reviews as well. I am happy with the current version of the manuscript as it stands.

No action taken.

Reviewer #3 (Remarks to the Author):

This new version takes into account all my comments as well as those made by the other reviewers.

No action taken.